# TimeXer: Empowering Transformers for Time Series Forecasting with Exogenous Variables

**Yuxuan Wang,**[*] **Haixu Wu,**[*] **Jiaxiang Dong, Guo Qin, Haoran Zhang,**
**Yong Liu, Yunzhong Qiu, Jianmin Wang, Mingsheng Long**[✉]
School of Software, BNRist, Tsinghua University, Beijing 100084, China
`{wangyuxu22,whx20,djx20,qinguo24,zhang-hr24,liuyong21,qiuyz24}@mails.tsinghua.edu.cn`
`{jimwang,mingsheng}@tsinghua.edu.cn`

## Abstract

Deep models have demonstrated remarkable performance in time series forecasting. However, due to the partially-observed nature of real-world applications, solely focusing on the target of interest, so-called *endogenous variables*, is usually insufficient to guarantee accurate forecasting. Notably, a system is often recorded into multiple variables, where the *exogenous variables* can provide valuable external information for endogenous variables. Thus, unlike well-established multivariate or univariate forecasting paradigms that either treat all the variables equally or ignore exogenous information, this paper focuses on a more practical setting: time series forecasting with exogenous variables. We propose a novel approach, **TimeXer**, to ingest external information to enhance the forecasting of endogenous variables. With deftly designed embedding layers, TimeXer empowers the canonical Transformer with the ability to reconcile endogenous and exogenous information, where patch-wise self-attention and variate-wise cross-attention are used simultaneously. Moreover, global endogenous tokens are learned to effectively bridge the causal information underlying exogenous series into endogenous temporal patches. Experimentally, TimeXer achieves consistent state-of-the-art performance on twelve real-world forecasting benchmarks and exhibits notable generality and scalability. Code is available at this repository: `https://github.com/thuml/TimeXer`.

## 1 Introduction

Time series forecasting is of pressing demand in real-world scenarios and have been widely used in various application domains, such as meteorology [38, 42], electricity [34], and transportation [27]. Thereof, forecasting with exogenous variables is a prevalent and indispensable forecasting paradigm since the variations within time series data are often influenced by external factors, such as economic indicators, demographic changes, and societal events. For example, electricity prices are highly dependent on the supply and demand of the market, and it is intrinsically impossible to predict future prices solely based on historical data. Incorporating external factors in terms of exogenous variables, as illustrated in Figure 1 (Left), allows for a more comprehensive understanding of the correlations and causalities among various variables, leading to better performance and interpretability.

From the perspective of time series modeling, exogenous variables are introduced to the forecaster for informative purposes and do not need to be predicted. The distinction between endogenous and exogenous variables poses unique challenges compared to existing multivariate forecasting methods. First, there are always multiple external factors that are illuminating to the prediction of the target series, which requires models to reconcile the discrepancy and dependency among *endogenous*

---

[*]Equal Contribution

38th Conference on Neural Information Processing Systems (NeurIPS 2024).

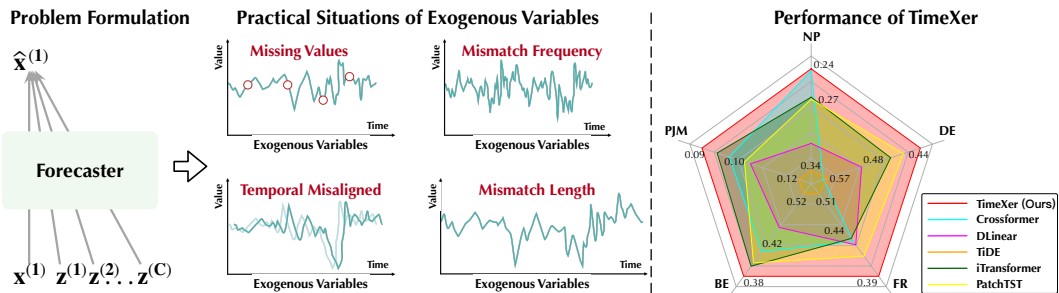

Figure 1: Left: The forecasting with exogenous variables paradigm includes inputs from multiple external variables as auxiliary information without the need for forecasting. Right: Model performance comparison on existing electricity price forecasting with exogenous variables benchmarks.

and *exogenous* variables. Regarding exogenous variables equally with endogenous ones will not only cause significant time and memory complexity but also involve unnecessary interactions from endogenous series to external information. Second, external factors may have a causal effect on endogenous series, so models are expected to reason about the systematic time lags among different variables. Moreover, as a practical forecasting paradigm applied extensively in real scenarios, it is essential for models to tackle irregular and heterogeneous exogenous series, including value missing, temporal misalignment, frequency mismatch, and length discrepancy as showcased in Figure 1(Left).

Despite the success of deep learning models in capturing intricate temporal dependencies in time series data, the incorporation of exogenous variables remains underexplored. A common practice to import them is adding or concatenating exogenous features to the endogenous ones. However, given the crucial role of exogenous variables in forecasting, it is imperative to incorporate them precisely and properly. Recent Transformers [32] have exhibited remarkable performance in time series forecasting due to their capability of capturing both temporal dependencies and multivariate correlations. Based on the working dimensions of the attention mechanism, existing Transformer-based works can be roughly divided into patch-oriented models and variate-oriented models. Patching is a basic module to preserve the semantic information underlying temporal variations. Therefore, the attention mechanism is applied over patches to unearth the intricate temporal patterns. Based on the channel independence assumption, PatchTST and follow-ups [28] are capable of capturing temporal dependencies but weak at capturing multivariate correlations. In contrast, variate-oriented models represented by iTransformer [23] successfully reason about interrelationships between variables by considering each variate of time series as a single token and applying attention over multiple variate tokens. Unfortunately, they lack the ability to capture internal temporal variations since the whole series is embedded into a coarse variate token by a temporal linear projection.

To enable accurate forecasting with exogenous variables in real-world scenarios, it is indispensable to capture both the intra-endogenous temporal dependencies and inter-series correlations between endogenous and exogenous variables. Based on the above observations, we believe that modeling the temporal-wise and variate-wise dependencies within time series data requires hierarchical representations at different levels. In this paper, we unleash the potential of the canonical Transformer without modifying any component, and propose a **Time** Series Transform**er** with e**X**ogenous variables (**TimeXer**). Technologically, we leverage representations and perform attention mechanisms at both patch and variate levels. First, the endogenous patch-level tokens are applied to capture temporal dependencies. Second, to tackle the arbitrarily irregular exogenous variables, TimeXer adopts their variate-level representations to seamlessly ingest the impact of external factors on endogenous ones. Third, inspired by Vision Transformers [10], we introduce learnable global tokens for each endogenous series to reflect the macroscopic information of the series, which interact simultaneously with patch-level endogenous tokens and variate-level exogenous tokens. Throughout this information pathway, the external information can be propagated effectively and selectively to corresponding endogenous patches. In summary, our contributions can be listed as follows.

- Motivated by the universality and importance of exogenous variables in time series forecasting, we empower the canonical Transformer to simultaneously modeling exogenous and endogenous variables without any architectural modifications.

- We propose a simple and general TimeXer model, which employs patch-level and variate-level representations respectively for endogenous and exogenous variables, with an en-

dogenous global token as a bridge in-between. With this design, TimeXer can capture intra-endogenous temporal dependencies and exogenous-to-endogenous correlations jointly.

- Extensive experiments on twelve datasets show that TimeXer can better utilize exogenous information to facilitate endogenous forecasting, in both univariate and multivariate settings.

## 2 Related Work

### 2.1 Transformer-based Time Series Forecaster

Motivated by the great success in the field of natural language processing [7] and computer vision [26], Transformers have garnered significant interest in time series data due to their ability to capture long-term temporal dependencies and complex multivariate correlations. Categorized based on the granularity of representation used in the attention mechanism, Transformer-based models can be divided into point-wise, patch-wise, and variate-wise. Due to the serial nature of time series, most previous works use a point-wise representation of time series data and apply attention mechanisms to capture the correlations among different time points. Therefore, many efficient Transformers [22, 37, 44, 45, 9] were proposed to reduce the complexity caused by point-wise modeling. Informer [44] designs a ProbSparse self-attention to reduce the quadratic complexity in time and memory. Autoformer [37] replaces canonical self-attention with Auto-correlation to discover the sub-series similarity within time series data. Pyraformer [22] develops a pyramidal attention module to capture both short- and long-temporal dependencies with linear time and space complexity.

Considering point-wise representations fall short in revealing local semantic information lies in the temporal variation, PatchTST [28] split time series data into subseries-level patches and then capture dependencies between patches. Pathformer [4] utilizes multi-scale patch representations and performs dual attention over these patches to capture global correlations and local details as temporal dependencies. Recent large-scale time series models [2, 6, 25, 46, 8] have widely included patch-level representation to learn the complex temporal patterns. Beyond capturing the patch-level temporal dependencies within one single series, recent approaches have endeavored to capture interdependencies among patches from different variables over time. Crossformer [43] introduces a Two-Stage Attention layer to efficiently capture the cross-time and cross-variate dependencies of each patch. Further expanding the receptive field, iTransformer [23] utilizes the global representation of the whole series and applies attention to these series-wise representations to capture multivariate correlations. Yet, as shown in Table 1, most of the existing Transformer-based approaches only focus on multivariate or univariate time series forecasting paradigms and do not conduct special designs for exogenous variables, which is different from the scenario we studied in this paper.

Table 1: Comparison of related methods with its forecasting capability. The character "." in the Transformers denotes the name of *former. The character ✧ indicates that the model can be applied to multivariate forecasting scenarios but not explicitly model the cross-variate dependency.

| Methods | **TimeXer** | iTran. [23] | PatchTST [28] | Cross. [43] | Auto. [37] | TFT [16] | NBEATSx [29] | TiDE [5] |
|---|---|---|---|---|---|---|---|---|
| Univariate | ✓ | ✗ | ✓ | ✗ | ✓ | ✗ | ✗ | ✗ |
| Multivariate | ✓ | ✓ | ✧ | ✓ | ✧ | ✗ | ✗ | ✓ |
| Exogenous | ✓ | ✗ | ✗ | ✗ | ✗ | ✓ | ✓ | ✓ |

### 2.2 Forecasting with Exogenous Variables

Time series forecasting with exogenous variables has been widely discussed in classical statistical methods. A vast majority of statistical methods have been extended to include exogenous variables as part of input. Extending the well-acknowledged ARIMA model, ARIMAX [35] and SARIMAX [31] incorporate the correlations between exogenous and endogenous variables along with the autoregression on endogenous variables. Although time series modeling methods have evolved from statistical to deep models, most of the existing deep models incorporating covariates, such as Temporal Fusion Transformer (TFT) [20], primarily focus on variable selection. Some approaches, including NBEATSx [29] and TiDE [5] contend that forecasting models are capable of accessing future values of exogenous variables during the prediction of endogenous variables. It is notable that previous models concatenate exogenous features with endogenous features at each time point and then map them to a latent space, necessitating the alignment of the endogenous and exogenous

series in time. However, time series in real-world applications often suffer from problems such as missing value and uneven sampling, which leads to significant challenges in modeling the effects of exogenous variables on endogenous variables. In contrast, TimeXer introduces external information to Transformer architecture through a deftly designed embedding strategy, which can effectively introduce the external information into patch-wise representations of endogenous variables, thereby being able to adapt to time-lagged or data-missing records.

## 3  TimeXer

In forecasting with exogenous variables, the endogenous series is the target to be predicted, while the exogenous series are covariates that provide valuable information to boost endogenous predictability.

**Problem Settings**  In forecasting with exogenous variables, we are given an endogenous time series $\mathbf{x}_{1:T} = \{x_1, x_2, ..., x_T\} \in \mathbb{R}^{T \times 1}$ and multiple exogenous series $\mathbf{z}_{1:T_{\mathrm{ex}}} = \{\mathbf{z}_{1:T_{\mathrm{ex}}}^{(1)}, \mathbf{z}_{1:T_{\mathrm{ex}}}^{(2)}, ..., \mathbf{z}_{1:T_{\mathrm{ex}}}^{(C)}\} \in \mathbb{R}^{T_{\mathrm{ex}} \times C}$. Here $x_i$ denotes the value at the $i$-th time point, $\mathbf{z}_{1:T_{\mathrm{ex}}}^{(i)}$ represents the $i$-th exogenous variable, and $C$ is the number of exogenous variables. In addition, $T$ and $T_{\mathrm{ex}}$ are the look-back window lengths of the endogenous and exogenous variables respectively. Noteworthily, any series that provides useful information for endogenous forecasting can be used as an exogenous variable, regardless of their look-back lengths, so we relax to the most flexible settings with $T_{\mathrm{ex}} \neq T$. The goal of forecasting model $\mathcal{F}_\theta$ parameterized by $\theta$ is to predict the future $S$ time steps $\widehat{\mathbf{x}} = \{x_{T+1}, x_{T+2}, ..., x_{T+S}\}$ based on both historical observations $\mathbf{x}_{1:T}$ and corresponding exogenous series $\mathbf{z}_{1:T_{\mathrm{ex}}}$:

$$\widehat{\mathbf{x}}_{T+1:T+S} = \mathcal{F}_\theta\left(\mathbf{x}_{1:T}, \mathbf{z}_{1:T_{\mathrm{ex}}}\right). \tag{1}$$

**Structure Overview**  As shown in Figure 2, the proposed TimeXer model repurposes the canonical Transformer without modifying any component, while endogenous and exogenous variables are manipulated by different embedding strategies. TimeXer adopts self-attention and cross-attention to capture temporal-wise and variate-wise dependencies respectively.

**Endogenous Embedding**  Most of the existing Transformer-based forecasting models embed each time point or a segment of time series as a temporal token and apply self-attention to learn temporal dependencies. To finely capture temporal variations within the endogenous variable, TimeXer adopts patch-wise representations. Concretely, the endogenous series is split into non-overlapping patches, and each patch is projected to a temporal token. Given the distinct roles of endogenous and exogenous variables in the prediction, TimeXer embeds them at *different* granularity. Therefore, directly combining endogenous tokens and exogenous tokens at different granularity will result in information misalignment. To address this, we introduce a learnable global token for each endogenous variable that serves as the macroscopic representation to interact with exogenous variables. This design helps bridge the causal information from the exogenous series to the endogenous temporal patches. The overall endogenous embedding is formally stated as:

$$\{\mathbf{s}_1, \mathbf{s}_2, ..., \mathbf{s}_N\} = \mathrm{Patchify}\left(\mathbf{x}\right),$$
$$\mathbf{P}_{\mathrm{en}} = \mathrm{PatchEmbed}\left(\mathbf{s}_1, \mathbf{s}_2, ..., \mathbf{s}_N\right), \tag{2}$$
$$\mathbf{G}_{\mathrm{en}} = \mathrm{Learnable}\left(\mathbf{x}\right).$$

Denote by $P$ the patch length, by $N = \lfloor \frac{T}{P} \rfloor$ the number of patches split from the endogenous series, and by $\mathbf{s}_i$ the $i$-th patch. $\mathrm{PatchEmbed}(\cdot)$ maps each length-$P$ patch, added by its position embedding, into a $D$-dimensional vector via a trainable linear projector. In all, $N$ patch-level temporal token embeddings $\mathbf{P}_{\mathrm{en}}$ and 1 series-level global token embedding $\mathbf{G}_{\mathrm{en}}$ are fed into the Transformer encoder.

**Exogenous Embedding**  The primary use of exogenous variables is to facilitate accurate forecasting of endogenous variables. We will show in Appendix B.3 that the interactions of different variables can be captured more naturally by variate-level representations, which are adaptive to arbitrary irregularities such as missing values, misaligned timestamps, different frequencies, or discrepant look-back lengths. In contrast, patch-level representations are overly fine-grained for exogenous variables, introducing not only significant computational complexity but also unnecessary noise information. These insights lead to a design that each exogenous series is embedded in a series-wise variate token, which is formalized as:

$$\mathbf{V}_{\mathrm{ex},i} = \mathrm{VariateEmbed}\left(\mathbf{z}^{(i)}\right), \; i \in \{1, \cdots, C\}. \tag{3}$$

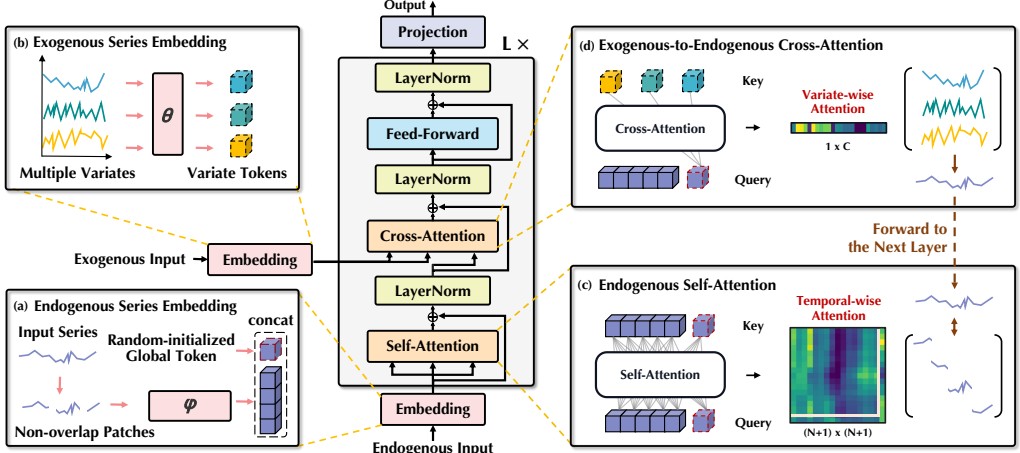

Figure 2: The schematic of TimeXer, which empowers time series forecasting with exogenous variables. (a) The endogenous embedding module yields multiple temporal token embeddings and one global token embedding for the endogenous variable. (b) The exogenous embedding module yields a variate token embedding for each exogenous variable. (c) Self-attention is applied simultaneously over the endogenous temporal tokens and the global token to capture patch-wise dependencies. (d) Cross-attention is applied over endogenous and exogenous variables to integrate external information.

Here VariateEmbed : $\mathbb{R}^{T_{\text{ex}}} \to \mathbb{R}^D$ is a trainable linear projector, $T_{\text{ex}}$ is the look-back length of exogenous series, and $\mathbf{V}_{\text{ex}} = \{\mathbf{V}_{\text{ex},i}\}_{i=1}^C$ is the set of representations for multiple exogenous series.

**Endogenous Self-Attention**  For accurate time series forecasting, it is vital to discover intrinsic temporal dependencies within the endogenous variable, as well as the interactions with the variate-level representations from exogenous variables. In addition to self-attention over endogenous temporal tokens (Patch-to-Patch), the learnable global token builds a *bridge* between endogenous and exogenous variables. Concretely, the global token plays an asymmetric role in cross-attention: (1) Patch-to-Global: the global token attends to temporal tokens for aggregating patch-level information across the entire series; (2) Global-to-Patch: each temporal token attends to the global token for receiving the variate-level correlations. This provides a comprehensive view of the temporal dependencies within the endogenous variable, as well as better interactions with the arbitrarily irregular exogenous variables. The attention mechanism can be formalized as follows:

$$\text{Patch-to-Patch:} \quad \widehat{\mathbf{P}}_{\text{en}}^{l,1} = \text{LayerNorm}\left(\mathbf{P}_{\text{en}}^l + \text{Self-Attention}\left(\mathbf{P}_{\text{en}}^l\right)\right),$$
$$\text{Global-to-Patch:} \quad \widehat{\mathbf{P}}_{\text{en}}^{l,2} = \text{LayerNorm}\left(\mathbf{P}_{\text{en}}^l + \text{Cross-Attention}\left(\mathbf{P}_{\text{en}}^l, \mathbf{G}_{\text{en}}^l\right)\right), \quad (4)$$
$$\text{Patch-to-Global:} \quad \widehat{\mathbf{G}}_{\text{en}}^l = \text{LayerNorm}\left(\mathbf{G}_{\text{en}}^l + \text{Cross-Attention}\left(\mathbf{G}_{\text{en}}^l, \mathbf{P}_{\text{en}}^l\right)\right).$$

The overall process can be simplified into an endogenous self-attention computation:

$$\widehat{\mathbf{P}}_{\text{en}}^l, \widehat{\mathbf{G}}_{\text{en}}^l = \text{LayerNorm}\left(\left[\mathbf{P}_{\text{en}}^l, \mathbf{G}_{\text{en}}^l\right] + \text{Self-Attention}\left(\left[\mathbf{P}_{\text{en}}^l, \mathbf{G}_{\text{en}}^l\right]\right)\right). \quad (5)$$

where $l \in \{0, \ldots, L-1\}$ denotes the $l$-th TimeXer block, and $\mathbf{P}_{\text{en}}^0 = \mathbf{P}_{\text{en}}$, $\mathbf{G}_{\text{en}}^0 = \mathbf{G}_{\text{en}}$. Here, $[\cdot, \cdot]$ denotes the concatenation of the patch-wise tokens and global token of the endogenous variable along the sequence dimension. By adopting a self-attention layer over the concatenated tokens $\left[\mathbf{P}_{\text{en}}^l, \mathbf{G}_{\text{en}}^l\right]$ of the endogenous series, TimeXer can capture temporal dependencies between patches and the relationships between each patch to the entire series simultaneously.

**Exogenous-to-Endogenous Cross-Attention**  Cross-attention has been widely used in multi-modal learning [17] to capture the adaptive token-wise dependencies between different modalities. In TimeXer, the cross-attention layer takes the endogenous variable as query and the exogenous variable as key and value to build the connections between the two types of variables. Since the exogenous variables are embedded into variate-level tokens, we use the learned global token of the endogenous variable to aggregate information from exogenous variables. The above process can be formalized as

$$\text{Variate-to-Global:} \quad \widehat{\mathbf{G}}_{\text{en}}^l = \text{LayerNorm}\left(\widehat{\mathbf{G}}_{\text{en}}^l + \text{Cross-Attention}\left(\widehat{\mathbf{G}}_{\text{en}}^l, \mathbf{V}_{\text{ex}}\right)\right). \quad (6)$$

Finally, all temporal tokens and the learnable global token will be transformed by the feedforward layer, which is formally stated as:

$$\mathbf{P}_{\text{en}}^{l+1} = \text{Feed-Forward}\left(\widehat{\mathbf{P}}_{\text{en}}^{l}\right), \mathbf{G}_{\text{en}}^{l+1} = \text{Feed-Forward}\left(\widehat{\mathbf{G}}_{\text{en}}^{l}\right), \tag{7}$$

where $l \in \{1, \ldots, L\}$. We write each Transformer block as $\mathbf{P}_{\text{en}}^{l+1}, \mathbf{G}_{\text{en}}^{l+1} = \text{TrmBlock}(\mathbf{P}_{\text{en}}^{l}, \mathbf{G}_{\text{en}}^{l})$.

**Forecasting Loss**   In time series forecasting with exogenous variables, the exogenous variables do not need to be predicted. So we generate the forecast $\widehat{\mathbf{x}}$ by applying a linear projection on the endogenous output embeddings $[\mathbf{P}_{\text{en}}^{L}, \mathbf{G}_{\text{en}}^{L}]$, a common practice in the encoder-only forecasters. We employ the squared loss (L2) to measure the discrepancy between the prediction and the ground truth:

$$\text{Loss} = \sum\nolimits_{i=1}^{S} \left\| \mathbf{x}_i - \widehat{\mathbf{x}}_i \right\|_2^2, \quad \text{where} \;\; \widehat{\mathbf{x}} = \text{Projection}\left([\mathbf{P}_{\text{en}}^{L}, \mathbf{G}_{\text{en}}^{L}]\right). \tag{8}$$

**Parallel Multivariate Forecasting**   Multivariate forecasting can be viewed as predicting each variable in the multivariate data, with the other variables treated as exogenous ones. So for each variable, the other variables are leveraged by TimeXer to facilitate more accurate and causal prediction. Our key discovery is that forecasting with exogenous variables can be a unified forecasting paradigm that generalizes straightforwardly to multivariate forecasting. By employing the *channel independence* mechanism, for each variable of the multivariate, it is treated as the endogenous one. Then TimeXer is applied in a parallel manner for all variables with shared self-attention and cross-attention layers.

## 4   Experiments

To verify the effectiveness and generality of TimeXer, we extensively experiment under two different time series paradigms, *i.e.* short-term forecasting with exogenous variables and long-term multivariate forecasting, on a diverse range of real-world time series datasets from different domains. We also conduct experiments on long-term forecasting with exogenous variables on the multivariate benchmark, which are presented in Appendix I.3.

**Datasets**   For short-term forecasting tasks, we include short-term electricity price forecasting datasets (EPF) [15], which is a real-world forecasting with exogenous various benchmarks derived from five major power market data spanning six years each. Each dataset contains electricity price as an endogenous variable and two influential exogenous variables in practice. Meanwhile, we adopt seven well-established public long-term multivariate forecasting benchmarks [33] to evaluate the performance of TimeXer in multivariate forecasting.

**Baselines**   We include nine state-of-the-art deep forecasting models, including Transformer-based models: iTransformer [23], PatchTST [28], Crossformer [43], Autoformer [37], CNN-based models: TimesNet [36], SCINet [21], and linear-based models: RLinear [19], DLinear [41], TiDE [5]. Notably, TiDE is a recently developed advanced forecaster specifically designed for exogenous variables.

**Implementation Details**   For short-term electricity price prediction, we follow the standard protocol of NBEATSx [29], where the input series length and prediction length are respectively set as 168 and 24. In addition, we set the patch length as 24 without overlapping. For long-term forecasting datasets, we uniformly use the patch length 16 and fix the length of the look-back series at 96, while the prediction length varies across four lengths $\{96, 192, 336, 720\}$.

### 4.1   Main Results

Comprehensive forecasting results for short-term and long-term forecasting are listed in Table 2 and Table 3. A lower MSE or MAE indicates better forecasting performance.

The short-term electricity price forecasting task is derived from real-world scenarios, and presents a unique challenge for the forecasting model for the endogenous variable has been shown to be highly correlated with two exogenous variables in the dataset. Since the interactions between different variables are crucial for this task, linear forecasters, including RLinear [19] and DLinear [41], fail to triumph over Transformer-based forecasters. Similar to TimeXer, Crossformer divides all input series into different segments and captures multivariate correlations over all segments; However,

it fails to outperform other baselines which indicates that modeling all variables at a granular level introduces unnecessary noise into the forecasting. Also designed for capturing cross-variate dependency, iTransformer neglects the temporal-wise attention module, indicating that there are still limitations in capturing temporal dependencies solely through linear projection. By contrast, our proposed TimeXer effectively integrates information from exogenous variables while capturing temporal dependencies of endogenous series. As shown in Table 2, TimeXer achieves consistent state-of-the-art performance on all five datasets, outperforming various baseline models.

Table 2: Full results of the short-term forecasting task on EPF dataset. We follow the standard protocol in short-term electricity price forecasting, where the input length and predict length are set to 168 and 24 respectively for all baselines. Avg means the average results from all five datasets.

| Model | TimeXer | | iTransformer | | RLinear | | PatchTST | | Crossformer | | TiDE | | TimesNet | | DLinear | | SCINet | | Autoformer | |
|---|---|---|---|---|---|---|---|---|---|---|---|---|---|---|---|---|---|---|---|---|---|
| Metric | MSE | MAE | MSE | MAE | MSE | MAE | MSE | MAE | MSE | MAE | MSE | MAE | MSE | MAE | MSE | MAE | MSE | MAE | MSE | MAE |
| NP | **0.236** | **0.268** | 0.265 | 0.300 | 0.335 | 0.340 | 0.267 | 0.284 | 0.240 | 0.285 | 0.335 | 0.340 | 0.250 | 0.289 | 0.309 | 0.321 | 0.373 | 0.368 | 0.402 | 0.398 |
| PJM | **0.093** | **0.192** | 0.097 | 0.197 | 0.124 | 0.229 | 0.106 | 0.209 | 0.101 | 0.199 | 0.124 | 0.228 | 0.097 | 0.195 | 0.108 | 0.215 | 0.143 | 0.259 | 0.168 | 0.267 |
| BE | **0.379** | **0.243** | 0.394 | 0.270 | 0.520 | 0.337 | 0.400 | 0.262 | 0.420 | 0.290 | 0.523 | 0.336 | 0.419 | 0.288 | 0.463 | 0.313 | 0.731 | 0.412 | 0.500 | 0.333 |
| FR | **0.385** | **0.208** | 0.439 | 0.233 | 0.507 | 0.290 | 0.411 | 0.220 | 0.434 | 0.208 | 0.510 | 0.290 | 0.431 | 0.234 | 0.429 | 0.260 | 0.855 | 0.384 | 0.519 | 0.295 |
| DE | **0.440** | **0.415** | 0.479 | 0.443 | 0.574 | 0.498 | 0.461 | 0.432 | 0.574 | 0.430 | 0.568 | 0.496 | 0.502 | 0.446 | 0.520 | 0.463 | 0.565 | 0.497 | 0.674 | 0.544 |
| AVG | **0.307** | **0.265** | 0.335 | 0.289 | 0.412 | 0.339 | 0.330 | 0.282 | 0.354 | 0.284 | 0.412 | 0.338 | 0.340 | 0.290 | 0.366 | 0.314 | 0.533 | 0.384 | 0.453 | 0.368 |

We also evaluate TimeXer on well-established public benchmarks for conventional multivariate long-term forecasting. As mentioned above, TimeXer has the ability to perform multivariate forecasting by employing the channel independence mechanism. We present the results averaged from all four prediction lengths in Table 3. It can be observed that TimeXer achieves consistent state-of-the-art performance on most of the datasets, highlighting its effectiveness and generality. In addition, since TimeXer is initially designed for exogenous variables, we also conduct vanilla forecasting with exogenous variables on these datasets by taking the last dimension of the multivariate data as endogenous series and others as exogenous variables. Detailed results are listed in Appendix I.3.

Table 3: Multivariate forecasting results. We compare extensive competitive models under different prediction lengths following the setting of iTransformer [23]. The look-back length $L$ is set to 96 for all baselines. Results are averaged from all prediction lengths $S = \{96, 192, 336, 720\}$.

| Model | TimeXer | | iTransformer | | RLinear | | PatchTST | | Crossformer | | TiDE | | TimesNet | | DLinear | | SCINet | | Autoformer | |
|---|---|---|---|---|---|---|---|---|---|---|---|---|---|---|---|---|---|---|---|---|---|
| Metric | MSE | MAE | MSE | MAE | MSE | MAE | MSE | MAE | MSE | MAE | MSE | MAE | MSE | MAE | MSE | MAE | MSE | MAE | MSE | MAE |
| ECL | **0.171** | **0.270** | 0.178 | 0.270 | 0.219 | 0.298 | 0.205 | 0.290 | 0.244 | 0.334 | 0.251 | 0.244 | 0.192 | 0.295 | 0.212 | 0.300 | 0.268 | 0.365 | 0.227 | 0.338 |
| Weather | **0.241** | **0.271** | 0.258 | 0.278 | 0.272 | 0.291 | 0.259 | 0.281 | 0.259 | 0.315 | 0.271 | 0.320 | 0.259 | 0.287 | 0.265 | 0.317 | 0.292 | 0.363 | 0.338 | 0.382 |
| ETTh1 | **0.437** | **0.437** | 0.454 | 0.447 | 0.446 | 0.434 | 0.469 | 0.454 | 0.529 | 0.522 | 0.541 | 0.507 | 0.458 | 0.450 | 0.456 | 0.452 | 0.747 | 0.647 | 0.496 | 0.487 |
| ETTh2 | **0.367** | **0.396** | 0.383 | 0.407 | 0.374 | 0.398 | 0.387 | 0.407 | 0.942 | 0.684 | 0.611 | 0.550 | 0.414 | 0.427 | 0.559 | 0.515 | 0.954 | 0.723 | 0.450 | 0.459 |
| ETTm1 | **0.382** | **0.397** | 0.407 | 0.410 | 0.414 | 0.407 | 0.387 | 0.400 | 0.512 | 0.496 | 0.419 | 0.419 | 0.400 | 0.406 | 0.403 | 0.407 | 0.485 | 0.481 | 0.588 | 0.517 |
| ETTm2 | **0.274** | **0.322** | 0.288 | 0.332 | 0.286 | 0.327 | 0.281 | 0.326 | 0.757 | 0.610 | 0.358 | 0.404 | 0.291 | 0.333 | 0.350 | 0.401 | 0.571 | 0.537 | 0.327 | 0.371 |
| Traffic | 0.466 | 0.287 | **0.428** | **0.282** | 0.626 | 0.378 | 0.481 | 0.304 | 0.550 | 0.304 | 0.760 | 0.473 | 0.620 | 0.336 | 0.625 | 0.383 | 0.804 | 0.509 | 0.628 | 0.379 |

## 4.2 Ablation Study

In TimeXer, three types of tokens are used to capture temporal-wise and variate-wise dependencies, including multiple patch-level temporal tokens, learnable global tokens of the endogenous variables, and multiple variate-level exogenous tokens. Besides, to incorporate the information from exogenous variables, TimeXer adopts a cross-attention layer to model the mutual relationship between different variables. To elaborate on the validity of TimeXer, we conducted detailed ablations covering both the embedding module and the inclusion of exogenous factors. Specifically, for the embedding design, we replace or remove existing components of the embedded vector from exogenous and endogenous variables respectively. Moreover, we keep the existing embedding design and replace the cross-attention by adding the variate token of exogenous variables to the variate token of endogenous variables or concatenating all the variate tokens and temporal tokens. As listed in Table 4, TimeXer exhibits superior performance compared to various architectural designs across all datasets.

Table 4: Ablation Results. *Ex.* and *En.* are abbreviations for Exogenous variable and Endogenous variable. *P*, *G* and *V* denote patch token, learnable global token, and variate token respectively.

| Design | | En. | Ex. | NP MSE | NP MAE | PJM MSE | PJM MAE | BE MSE | BE MAE | FR MSE | FR MAE | DE MSE | DE MAE | AVG MSE | AVG MAE |
|---|---|---|---|---|---|---|---|---|---|---|---|---|---|---|---|
| Cross | **Ours** | P+G | V | **0.236** | **0.268** | **0.093** | **0.192** | 0.379 | **0.243** | **0.385** | **0.208** | **0.440** | **0.415** | **0.307** | **0.265** |
| | Replace | P+G | P | 0.237 | 0.269 | 0.101 | 0.196 | **0.376** | 0.246 | 0.390 | 0.206 | 0.457 | 0.422 | 0.312 | 0.268 |
| | Remove | P | V | 0.239 | 0.273 | 0.106 | 0.200 | 0.381 | 0.260 | 0.393 | 0.208 | 0.468 | 0.425 | 0.316 | 0.273 |
| Add | | P+G | V | 0.247 | 0.272 | 0.125 | 0.206 | 0.387 | 0.247 | 0.404 | 0.209 | 0.483 | 0.430 | 0.329 | 0.273 |
| Concatenate | | P+G | V | 0.237 | 0.266 | 0.098 | 0.196 | 0.383 | 0.255 | 0.390 | 0.209 | 0.450 | 0.423 | 0.312 | 0.270 |

## 4.3 TimeXer Generality

### 4.3.1 Practical Situations

**Increasing Look-back Length** Theoretically, the forecasting performance of the model could potentially benefit from increasing the look-back length of time series, as a longer historical context encompasses more comprehensive information. However, the attention will be distracted when the look-back length becomes excessively long. In TimeXer, we use the variate-level representation of exogenous variables which allows for the misalignment between endogenous and exogenous variables. This is particularly valuable in real-world scenarios where the time series data may be collected from a newly introduced sensor that has limited historical data. Therefore, we conducted three different experimental settings to assess the generality of TimeXer by increasing the length of either the endogenous or exogenous series, which include "Fix Endogenous and increase Exogenous", "Increase Endogenous and Fix exogenous", and "Increase Endogenous and Exogenous". Results shown in Figure 3 reveal that TimeXer can be adapted to situations where the look-back of endogenous and exogenous are mismatched. Moreover, extending the look-back length indeed yields improvements in forecasting performance. Compared to enlarging the historical exogenous series, increasing the look-back length of the endogenous series brings greater benefits to the model, and the performance is further improved with both increases.

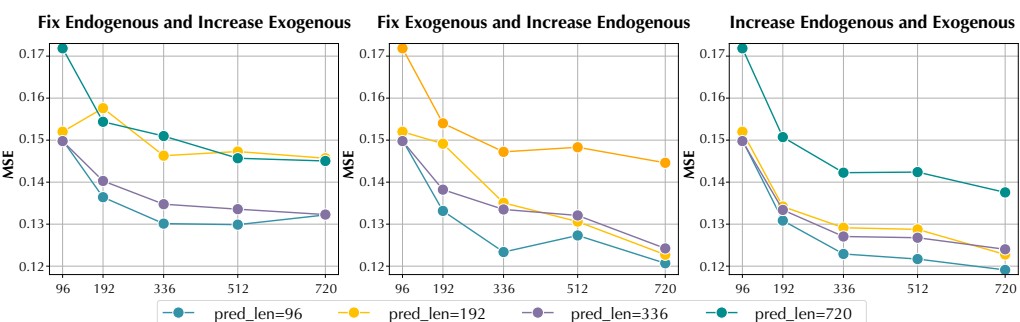

Figure 3: Performance with the enlarged look-back length varying from $\{96, 192, 336, 512, 720\}$. Different styles of lines represent different prediction lengths. In most cases, the forecasting performance benefits from enlarged look-back lengths of both endogenous and exogenous series.

**Missing Values** To further verify the generalizability of TimeXer in complex real-world scenarios, we conduct experiments in scenarios where the historical information of time series is missing. Specifically, for both exogenous and endogenous series, we adopt two strategies to evaluate TimeXer's adaptability to series with missing historical information: (1) **Zeros**: filling the whole series with the scalar value 0. (2) **Random**: substituting the whole series with random values from a uniform distribution on the interval $[0, 1)$. As shown in Table 5, the forecasting results deteriorate when exogenous variables are replaced with meaningless noise, indicating that the model's performance benefits from the inclusion of informative exogenous variables. Interestingly, neither using zero-filled exogenous series nor employing exogenous series with random numbers results in a significant

Table 5: Model performance under missing values. *Zeros* and *Random* represent the cases that the corresponding series is set as zeros or random values respectively.

| Variate | Strategies | NP MSE | NP MAE | PJM MSE | PJM MAE | BE MSE | BE MAE | FR MSE | FR MAE | DE MSE | DE MAE | AVG MSE | AVG MAE |
|---------|------------|------|------|------|------|------|------|------|------|------|------|------|------|
| Endogenous | Zeros | 2.954 | 1.396 | 0.188 | 0.288 | 0.930 | 0.664 | 0.781 | 0.534 | 0.774 | 0.559 | 1.125 | 0.688 |
| Endogenous | Random | 3.140 | 1.450 | 0.233 | 0.325 | 0.926 | 0.667 | 0.761 | 0.527 | 0.692 | 0.533 | 1.150 | 0.701 |
| Exogenous | Zeros | 0.257 | 0.278 | 0.108 | 0.210 | 0.400 | 0.254 | 0.416 | 0.214 | 0.471 | 0.430 | 0.330 | 0.277 |
| Exogenous | Random | 0.258 | 0.280 | 0.110 | 0.212 | 0.399 | 0.253 | 0.424 | 0.221 | 0.475 | 0.432 | 0.333 | 0.280 |
| TimeXer | | **0.236** | **0.268** | **0.093** | **0.192** | 0.379 | **0.243** | **0.385** | **0.208** | **0.440** | **0.415** | **0.307** | **0.265** |

decline in model performance. This robustness can be attributed to TimeXer's design, which uses two attention layers to model endogenous temporal dependencies and the multivariate correlations between endogenous and exogenous variables respectively. This architecture allows endogenous temporal representations to dominate the predictions, ensuring consistent performance even in the presence of uninformative exogenous data. Consequently, it can be observed that when the endogenous series is replaced with meaningless zeros or random values, rendering the time series unpredictable, there is a significant decline in model performance. This underscores that TimeXer's performance is closely tied to the quality of endogenous series, deteriorating markedly when the historical information is severely limited.

### 4.3.2 Scalability

Since recent Transformer-based forecasters have demonstrated promising scalability, leading to the success of Large Time Series Models, we explore the scalability of TimeXer on large-scale time series data. Specifically, we build a large-scale weather dataset for forecasting with exogenous variables. The endogenous series is the hourly temperature of 3,850 stations worldwide, spanning from January 1, 2019, to December 31, 2020, which can be downloaded from the National Centers for Environmental Information (NCEI) [1] and has been well-processed by [38]. Further, we utilize meteorological indicators of corresponding adjacent areas from ERA5 [11] as exogenous variables, which is with a sampling interval of 3 hours. The adjacent area is defined as the 3x3 grid centered on the endogenous weather station, with four meteorological variables per grid cell, totaling 36 exogenous variables. We set the historical horizon of endogenous and exogenous to be 7 days to predict the endogenous variable for the next 3 days. Noteworthily, this is a complex forecasting scenario as we aforementioned where the frequencies of endogenous and exogenous are different. We choose existing state-of-the-art multivariate forecasters as baselines and use identical hidden dimensions and batch sizes for a fair comparison. Since baseline forecasters cannot handle mismatched series, we interpolate the exogenous series into hourly data using the nearest values. Figure 4 demonstrates that TimeXer surpasses other baselines, verifying its capability to handle large-scale forecasting tasks.

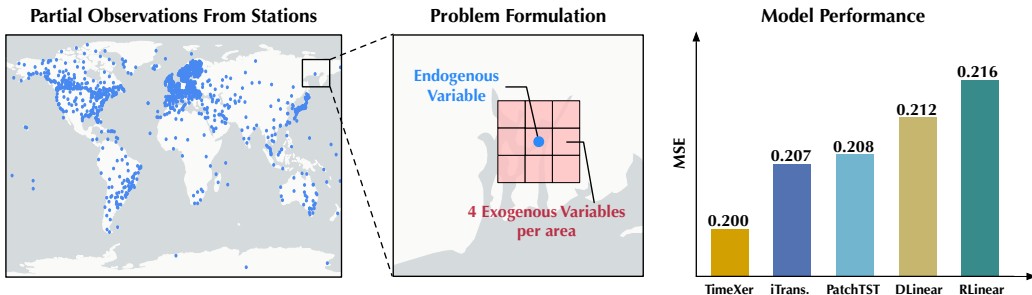

Figure 4: Forecasting performance on large-scale time series datasets. Left: Illustration of the forecasting scenario. The endogenous is the temperature collected from weather stations, and the exogenous variables are meteorological indicators from the surrounding 3x3 grids including the weather station. Each area contains four types of information, namely, temperature, pressure, u- and v- components of wind. Right: TimeXer outperforms other advanced forecasters.

## 4.4 Model Analysis

**Variate-wise Correlations**    TimeXer adopts cross-attention between the global endogenous token and variable-level exogenous tokens to capture the multivariate correlation, enhancing the interpretability of the learned attention map. To validate the rationale behind attention on variate tokens, we visualize the learned attention map alongside the time series of the highest and lowest attention scores. As illustrated in Figure 5 (Left), the case study on the Weather dataset reveals a notable distinction in the attention maps of endogenous variables with different exogenous variables. This demonstrates that TimeXer has the ability to distinguish between exogenous variables, allocating greater attention to those that are most informative for prediction, thereby resulting in a more focused and interpretable attention map. Additionally, it is observed that exogenous series exhibiting similar shapes to the endogenous series tend to receive more attention. This phenomenon may arise because time series with analogous shapes often share temporal features, leading to higher similarity scores. Consequently, the exogenous series most prominently highlighted by the attention mechanism may intuitively resemble the endogenous variable. Furthermore, physical interpretations for the visualized are provided. For the endogenous variable CO2-Concentration, there is indeed a strong correlation between it and Air Density, while the Maximum Wind Velocity has a relatively minor impact, which validates the effectiveness of TimeXer.

**Model Efficiency**    To evaluate the efficiency of TimeXer, we evaluate the training time and memory footprint of TimeXer on forecasting with exogenous variables compared with six baseline models with the identical hidden dimension and batch size for a fair comparison. We present the results on the ECL dataset with 320 exogenous variables in Figure 5 (Right). It is notable that when faced with numerous variables TimeXer exhibits its advantage by outperforming iTransformer in terms of memory footprint. Notably, iTransformer embeds each variate series into one token and applies a self-attention mechanism among all variate tokens, whether endogenous or exogenous. Although this design can keep refining the learned variate token in multiple layers, it does cause more complexity. As for TimeXer, exogenous variables will be embedded to variate tokens at the beginning, which will be shared in all layers and interact with the endogenous global token by cross-attention. Thus, TimeXer omits the interaction among learned exogenous variate tokens, resulting in favorable efficiency. We provide a comprehensive theoretical analysis of the model efficiency in Appendix E.

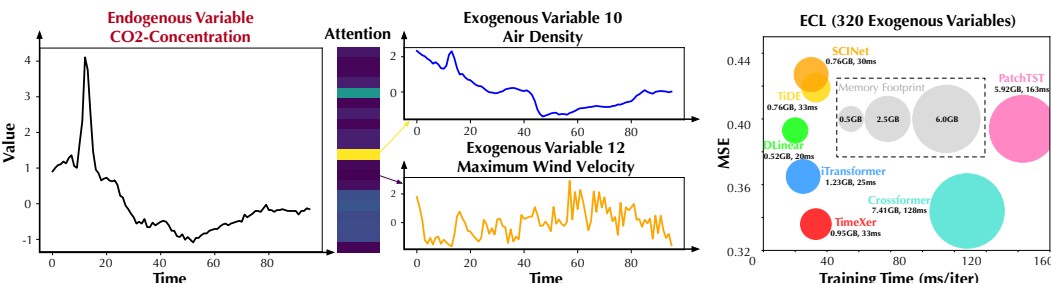

Figure 5: Model analysis of TimeXer. Left: Visualization of learned attention map and the endogenous time series and exogenous time series with highest and lowest attention scores. Right: Model efficiency comparison under the forecasting with exogenous variables paradigm on the ECL dataset.

## 5    Conclusion

Considering the prevalence of exogenous variables in real-world forecasting scenarios, we empower the canonical Transformer architecture with the ability to incorporate exogenous information without architectural modifications. Technologically, TimeXer revisits the attention mechanism in a per-patch-per-variate manner to capture both endogenous temporal dependencies and multivariate correlations between endogenous and exogenous variables. With a deftly designed global token, our proposed TimeXer is able to reconcile variables of different purposes. Experimental results demonstrate that our proposed TimeXer effectively ingests exogenous information to facilitate the prediction of endogenous series, in both univariate and multivariate settings. Besides, TimeXer has shown the potential scalability and promising abilities to address complex real-world forecasting scenarios, including challenges such as value missing, temporal misalignment, or series heterogeneity.

## Acknowledgments

This work was supported by the Ministry of Industry and Information Technology of China.

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

## A Implementation Details

### A.1 Dataset Descriptions

We conduct long-term forecasting experiments on 7 real-world datasets to evaluate the performance of our proposed TimeXer, including: (1) **ECL** [18] includes hourly electricity consumption data from 321 clients. We take the electricity consumption of the last client as an endogenous variable and other clients as exogenous variables. (2) **Weather** [44] records 21 meteorological factors collected every 10 minutes from the Weather Station of the Max Planck Biogeochemistry Institute in 2020. In our experiment, we use the Wet Bulb factor as the endogenous variable to be predicted and the other indicators as exogenous variables. (3) **ETT** [44] contains four subsets where ETTh1 and ETTh2 are hourly recorded, and ETTm1 and ETTm2 are recorded every 15 minutes. The endogenous variable is the oil temperature and the exogenous variables are 6 power load features. (4) **Traffic** [36] records hourly road occupancy rates measured by 862 sensors of San Francisco Bay area freeways. We take the measurement of the last sensor as an endogenous variable and others as exogenous variables.

In addition to the public multivariate time series datasets, we perform short-term forecasting on the electricity price forecasting datasets [15], which contains five datasets representing five different day-ahead electricity markets spanning six years each. Here are the descriptions of the datasets: (1) **NP** represents The Nord Pool electricity market, recording the hourly electricity price, and corresponding grid load and wind power forecast from 2013-01-01 to 2018-12-24. (2) **PJM** represents the Pennsylvania-New Jersey-Maryland market, which contains the zonal electricity price in the Commonwealth Edison (COMED), and corresponding System load and COMED load forecast from 2013-01-01 to 2018-12-24. (3) **BE** represents Belgium's electricity market, recording the hourly electricity price, load forecast in Belgium, and generation forecast in France from 2011-01-09 to 2016-12-31. (4) **FR** represents the electricity market in France, recording the hourly prices, and corresponding load and generation forecast from 2012-01-09 to 2017-12-31. (5) **DE** represents the German electricity market, recording the hourly prices, the zonal load forecast in the TSO Amprion zone, and the wind and solar generation forecasts from 2012-01-09 to 2017-12-31.

Table 6: Dataset descriptions. *Ex.* and *En.* are abbreviations for the Exogenous variable and Endogenous variable, respectively. The dataset size is organized in (Train, Validation, Test)

| Dataset | #Num | Ex. Descriptions | En. Descriptions | Sampling Frequency | Dataset Size |
|---|---|---|---|---|---|
| Electricity | 320 | Electricity Consumption | Electricity Consumption | 1 Hour | (18317, 2633, 5261) |
| Weather | 20 | Climate Feature | CO2-Concentration | 10 Minutes | (36792, 5271, 10540) |
| ETTh | 6 | Power Load Feature | Oil Temperature | 1 Hour | (8545, 2881, 2881) |
| ETTm | 6 | Power Load Feature | Oil Temperature | 15 Minutes | (34465, 11521, 11521) |
| Traffic | 861 | Road Occupancy Rates | Road Occupancy Rates | 1 Hour | (12185, 1757, 3509) |
| NP | 2 | Grid Load, Wind Power | Nord Pool Electricity Price | 1 Hour | (36500, 5219, 10460) |
| PJM | 2 | System Load, SyZonal COMED load | Pennsylvania-New Jersey-Maryland Electricity Price | 1 Hour | (36500, 5219, 10460) |
| BE | 2 | Generation, System Load | Belgium's Electricity Price | 1 Hour | (36500, 5219, 10460) |
| FR | 2 | Generation, System Load | France's Electricity Price | 1 Hour | (36500, 5219, 10460) |
| DE | 2 | Wind power, Ampirion zonal load | German's Electricity Price | 1 Hour | (36500, 5219, 10460) |

### A.2 Implementation Details

All the experiments are implemented in PyTorch [30] and conducted on a single NVIDIA 4090 24GB GPU. We utilize ADAM [13] with an initial learning rate $10^{-4}$ and L2 loss for the model optimization. The training process is fixed to 10 epochs with an early stopping. We set the number of TimeXer blocks in our proposed model $L \in \{1, 2, 3\}$. The dimension of series representations $d_{model}$ is searched from $\{128, 256, 512\}$. The patch length is uniformly set to 16 for long-term forecasts and 24 for short-term forecasts. We reproduced the compared baseline models based on the benchmark of TimesNet [36] Repository.

## B Ablation Study

### B.1 Using Overlapped Patch

In this paper, the proposed TimeXer adopts patch-wise representations of endogenous series via splitting the series into non-overlapping patches. Here we conduct ablation study on the patching

method. Following PatchTST [28], we set the patch length to 24, consistent with TimeXer, and the stride is set to 12, to generate a sequence of overlapped patches. Compared to the overlapping method, TimeXer has the lowest complexity while having the optimal performance. It is also notable that not only in NLP and CV, contemporary time series approaches also use non-overlapping patches. This preference might stem from the limited redundancy present in time series data, as excessive overlap can result in a smoothed representation for each patch, consequently failing to capture correct temporal dependencies.

Table 7: Model performance with overlapped patches.

| Design | NP | | PJM | | BE | | FR | | DE | | AVG | |
|---|---|---|---|---|---|---|---|---|---|---|---|---|
| | MSE | MAE | MSE | MAE | MSE | MAE | MSE | MAE | MSE | MAE | MSE | MAE |
| TimeXer | **0.236** | **0.268** | **0.093** | **0.192** | 0.379 | **0.243** | **0.385** | **0.208** | **0.440** | **0.415** | **0.307** | **0.265** |
| TimeXer-overlap | 0.240 | 0.267 | 0.095 | 0.194 | 0.383 | 0.248 | 0.409 | 0.214 | 0.453 | 0.419 | 0.316 | 0.269 |

## B.2 Varying Patch Length

In this section, we experiment with the effect of patch lengths on the forecasting performance. We fix the look-back window to 96 and vary the patch lengths $P \in \{2, 4, 6, 8, 12, 24\}$. We conduct experiments on five short-term electricity price forecasting datasets with consistent parameters except patch-length, and the results are shown in Figure 6. It can be seen that the average prediction performance does not vary dramatically with different patch lengths, which indicates that our model is robust to the patch length hyperparameter. Notably, the model generally performs lower when the patch length is small, which is probably because small patches are not enough to represent the semantic information in time series data.

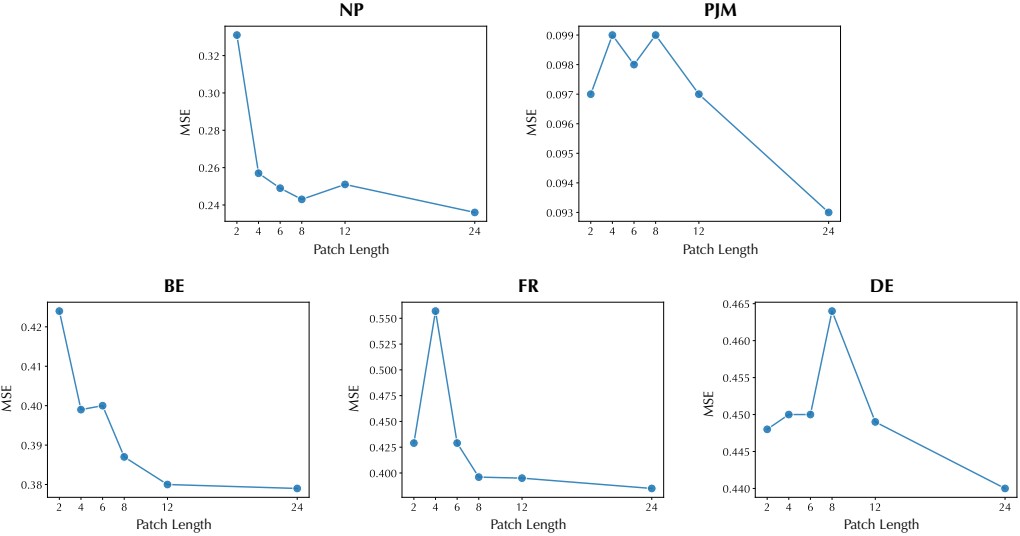

Figure 6: Hyper-parameter sensitivity analysis of TimeXer on short-term forecasting benchmarks.

## B.3 Alternative Embedding Approach

In our above-mentioned analysis, we apply representation from different level to capture the temporal dependencies and multivariate correlations. For the endogenous variable, we apply patch-level temporal token and a learnable global token to capture both temporal dependencies and cross-attention between different variables. And for exogenous variables, we utilize variate-level representation directly embedded from the whole series. To verify the rationality of our proposed architecture, we modify the embedding design and inclusion of exogenous variables. In this section, we provide

Table 8: Ablation study on long-term forecast.

| Design | Endogenous | Exogenous | Horizon | ETTh1 | | ETTm1 | | Traffic | |
|---|---|---|---|---|---|---|---|---|---|
| | | | | MSE | MAE | MSE | MAE | MSE | MAE |
| Ours | Patch+Global | Variate | 96 | 0.056 | 0.179 | 0.028 | 0.125 | 0.150 | 0.225 |
| | | | 192 | 0.071 | 0.205 | 0.043 | 0.158 | 0.152 | 0.228 |
| | | | 336 | 0.080 | 0.222 | 0.057 | 0.185 | 0.150 | 0.231 |
| | | | 720 | 0.084 | 0.229 | 0.079 | 0.217 | 0.172 | 0.253 |
| | | | Avg | **0.073** | **0.209** | **0.052** | **0.171** | 0.156 | 0.234 |
| Replace | Patch+Global | Patch | 96 | 0.057 | 0.182 | 0.028 | 0.125 | 0.156 | 0.232 |
| | | | 192 | 0.072 | 0.207 | 0.043 | 0.158 | 0.154 | 0.232 |
| | | | 336 | 0.083 | 0.226 | 0.058 | 0.186 | 0.154 | 0.238 |
| | | | 720 | 0.088 | 0.234 | 0.079 | 0.216 | 0.175 | 0.258 |
| | | | Avg | 0.075 | 0.212 | 0.052 | 0.171 | 0.160 | 0.240 |
| Remove | Patch | Variate | 96 | 0.058 | 0.183 | 0.028 | 0.126 | 0.153 | 0.229 |
| | | | 192 | 0.072 | 0.207 | 0.044 | 0.159 | 0.152 | 0.231 |
| | | | 336 | 0.081 | 0.222 | 0.058 | 0.187 | 0.152 | 0.235 |
| | | | 720 | 0.092 | 0.239 | 0.080 | 0.217 | 0.175 | 0.258 |
| | | | Avg | 0.076 | 0.213 | 0.052 | 0.172 | 0.160 | 0.240 |
| Add | Patch+Global | Variate | 96 | 0.058 | 0.183 | 0.029 | 0.128 | 0.168 | 0.240 |
| | | | 192 | 0.074 | 0.208 | 0.044 | 0.161 | 0.169 | 0.244 |
| | | | 336 | 0.085 | 0.227 | 0.060 | 0.189 | 0.165 | 0.246 |
| | | | 720 | 0.093 | 0.240 | 0.083 | 0.221 | 0.184 | 0.266 |
| | | | Avg | 0.078 | 0.214 | 0.054 | 0.175 | 0.171 | 0.249 |
| Concatenate | Patch+Global | Variate | 96 | 0.063 | 0.196 | 0.028 | 0.125 | 0.146 | 0.222 |
| | | | 192 | 0.074 | 0.210 | 0.045 | 0.162 | 0.148 | 0.225 |
| | | | 336 | 0.094 | 0.242 | 0.061 | 0.190 | 0.147 | 0.227 |
| | | | 720 | 0.088 | 0.233 | 0.080 | 0.219 | 0.156 | 0.239 |
| | | | Avg | 0.080 | 0.220 | 0.053 | 0.174 | **0.149** | **0.228** |

ablation results on long-term forecasting datasets. As shown in Table 8, among various architectural designs, TimeXer generally exhibits optimal performance. Notably, when replacing variate embedding of exogenous variables with patch embedding, prediction accuracy declines. This observation suggests that a series-wise representation of the exogenous variables is more advantageous for predicting the endogenous variable. However, it is important to note that applying patch-wise representations for all exogenous variables would significantly increase computational complexity. In contrast, TimeXer offers a more efficient and effective design. Nevertheless, we notice that on the Traffic dataset when concatenating the variate-level exogenous tokens with those endogenous tokens and performing a Self-Attention over them outperforms our proposed design. Since TimeXer only adopts Cross-Attention between endogenous and exogenous variables, the main distinction between these two designs is whether there is attention within multiple exogenous tokens. These ablation results indicate that the interaction within exogenous variables can also be viewed as external factors that may facilitate the prediction. However, this kind of correlation is not valid in all cases. As listed in Table 8, there is no improvement in ETTh1 and ETTm1 datasets but decreased.

## C TimeXer Generality under Missing Exogenous Values

Real-world time series encounter problems such as missing data that result in low-quality data. In this section, we use random masking to replicate these situations and further explore the forecasting

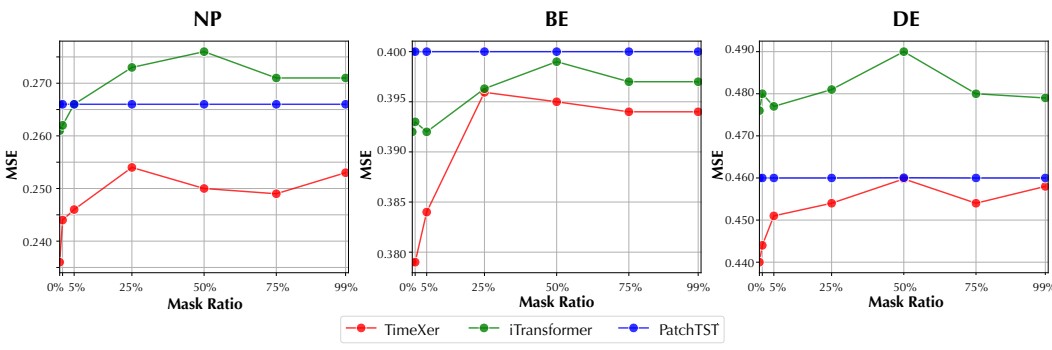

Figure 7: Forecasting performance with the masked exogenous series on three EPF datasets, simulating the missing values scenario.

performance when fed low-quality data. Previous works [9] have demonstrated that the semantic information of time series lies in temporal variation, We use complete, high-quality historical data for the endogenous variables and progressively reduce the quality of the data for the exogenous variables by increasing the masking ratio from 0% (i.e., using complete historical data for the exogenous variables) to 99% as shown in Figure 7. It can be observed that with the decrease in the quality of the exogenous series, the forecasting performance of the model also decreases. Notably, our model maintains a competitive performance when the exogenous series are masked by a small amount, which indicates that our proposed TimeXer is capable of supporting low-quality data scenarios.

## D    Increasing the Look-back Length

In the main text, we have already evaluated the forecasting performance with the increase of the look-back length of endogenous or exogenous series under forecasting with the exogenous variables paradigm. Here, we present results on multivariate forecasting with increased historical length. Results shown in Figure 8 (Right) indicate that TimeXer can benefit from the extended look-back length for a better performance.

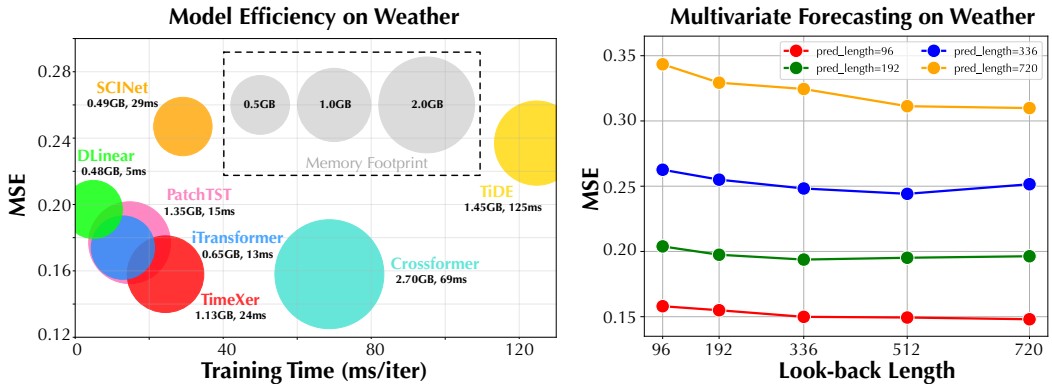

Figure 8: Model analysis on Weather dataset under multivariate forecasting paradigm. Left: Model efficiency comparison under input-96-predict-96. Right: Forecasting performance with an enlarged look-back length $T \in \{96, 192, 336, 512, 720\}$.

## E    Model Efficiency

In this section, we first provide a theoretical analysis of the computational complexity of TimeXer with other advanced Transformer-based forecasters. Suppose the look-back length and prediction length are $T$ and $S$ respectively, $C$ is the number of exogenous variables, and $P$ is the length of the patch. As we presented in Figure 5 in the main text, our proposed TimeXer has a clear advantage when the number of exogenous variables $C$ is large and achieves a favorable balance between modeling fineness and efficiency. This mainly benefits from the Cross-Attention to obtain a $O(C)$ complexity in variate dimensions, whereas iTransformer will increase to $O(C^2)$.

Beyond vanilla forecasting with exogenous variables, we also compare the efficiency under the multivariate forecasting paradigm on the Weather dataset when using historical 96-time steps to predict future 96-time steps. The result in Figure 8 clearly demonstrates that TimeXer shares a similar performance with Crossformer, but significantly outperforms in terms of training time and memory usage. Moreover, the efficiency of TimeXer is close to PatchTST, which is because the parallel multivariate forecasting is achieved by the channel independence mechanism on the self-attention over endogenous variables, which introduces a quadratic complexity $O((\frac{L}{P}+1)^2)$. Notably, this quadratic complexity can be reduced through (1) *Adapt the patch length*: The computational complexity is highly related to the patch length. (2) *Incorporate advanced attention module*: Since TimeXer does not modify any component in the Transformer, which results in a quadratic complexity. By replacing the full attention with the advanced attention module, such as linear attention, the complexity can decrease to $O((\frac{L}{P})+C)$.

## F    Representation Analysis

To evaluate the performance of TimeXer from the perspective of representation, we adopt centered kernel alignment (CKA) similarity [14] in this section. Previous works [36, 9] reveal that for low-level time series tasks including forecasting, there is a great similarity among representations from the different layers and a higher similarity corresponds to a better performance. Technologically, we calculate the CKA between the output features of the first and the last block obtained from TimeXer as well as other baselines. In particular, since iTransformer [23] is a multivariate forecaster that treats all variables equally, we provide the similarity corresponding to the endogenous variate in addition to the global-view CKA. As shown in Figure 9, the representations from the first and the last layers of TimeXer enjoy great similarities, verifying that TimeXer can learn the appropriate representations for the prediction. It is notable that iTransformer does not distinguish between endogenous and exogenous variables and the output of the model contains representations of all variables. However, the result of the CKA analysis shows that despite the high similarity of the series representations of all variables, the representation of the endogenous variables was not well learned. This result also suggests that directly applying a multivariate model to perform forecasting with exogenous variables introduces unnecessary noise into the model thus interfering with its forecasting performance.

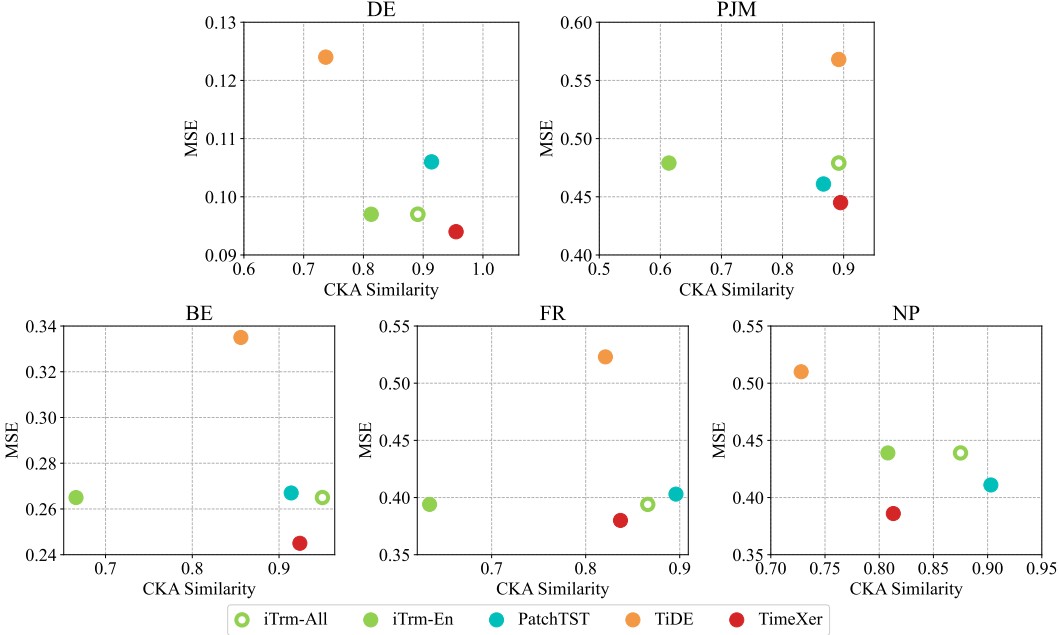

Figure 9: Series Representation Analysis on three EPF datasets. *iTrm-All* denotes the series representation of all variables learned by iTransformer, *iTrm-En* is the learned series representation of the endogenous variable.

## G    Discussion

We find that the multivariate forecasting results on Traffic datasets in Table 3 are different from the other datasets. Noteworthily, the mean absolute error (MAE) result of TimeXer is close to iTransformer, which is the state-of-the-art model on the Traffic dataset, but there is still a large margin in mean squared error (MSE). This unexpected result prompted us to further investigate. As a result, we provide a visualization of the forecasting results from TimeXer and iTransformer in Figure 10. Upon analysis, we observed that while TimeXer effectively predicted the overall trends of the future horizon, it struggled to accurately forecast the numerical value of future spikes. Subsequently, the squared calculation in MSE will amplify this error, resulting in a drastic difference compared to MAE. To further explore this performance, we also visualize the forecasting results given by PatchTST. As illustrated in Figure 10, we discover that PatchTST performs quite similarly to TimeXer. The commonality of these two models is that they both utilize patch-level representation to model the

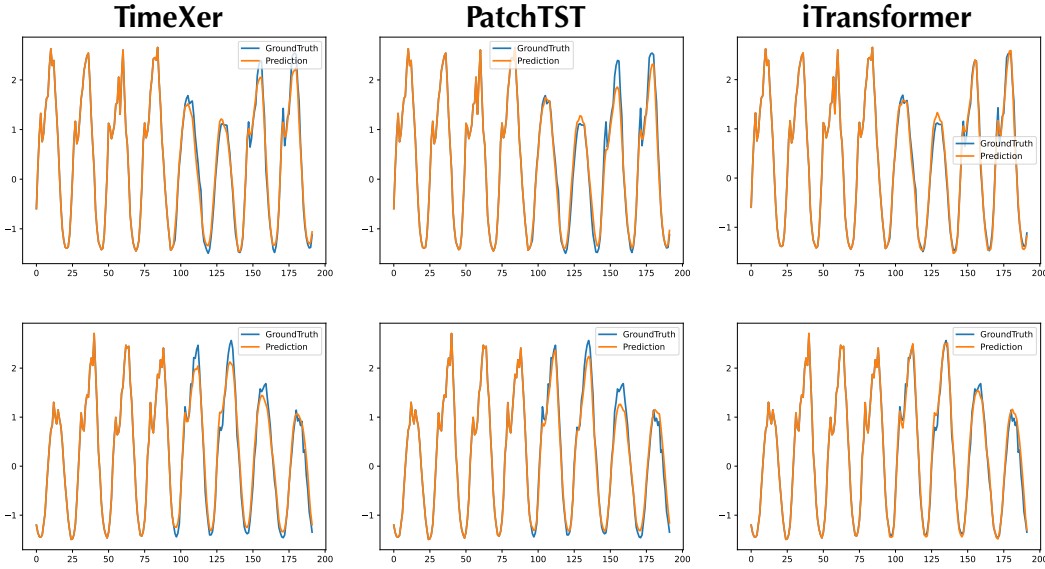

Figure 10: Multivariate forecasting showcase on Traffic dataset.

temporal dependencies, which we think might be the answer to the performance decrease. The utilization of patch-level representation in TimeXer and PatchTST contributes to their ability to capture temporal dependencies and contributes to the accurate prediction of the trends. Conversely, iTransformer focuses on the variate-wise correlation while the temporal correlation is only obtained through a linear projection. In our proposed TimeXer, we employ a linear projection over all the endogenous tokens, including multiple patch-level temporal and only one global variate-level token. Therefore it can be inferred that the excessive number of patches may make the prediction pay more attention to the overall trend change and fail to predict the precise value of changing points. Based on the above analysis, we believe that to alleviate this problem, we need to address the imbalance problem of multiple temporal tokens and only one global token, which can be solved by increasing the patch length or learnable tokens.

# H    Showcase

## H.1    Intuitive Showcases for Exogenous Variable Utilization

To enhance the understanding of the role of exogenous variables in prediction, we visually present the prediction results in two distinct scenarios: with and without exogenous variables. In the case where exogenous variables are incorporated, we introduce a special scenario in which the model has access to predictions of these exogenous variables. This is a practical scenario for the EPF datasets where the exogenous variables are the day-ahead predictions of the source generation. Additionally, we add an extreme case where there is no historical information on endogenous series to explore whether TimeXer can learn from exogenous variables in scenarios where only external information is available. By exploring these varied scenarios, we aim to provide a comprehensive analysis of how exogenous variables influence the model's forecasting performance. As illustrated in Figure 11, we can observe that removing either endogenous or exogenous variables leads to poorer predictions from the model. This indicates that both types of variables play a crucial role in enhancing the forecaster's performance. Notably, when the model has access to future predicted values of the exogenous variables, the performance achieves the best. This finding underscores the importance of incorporating external information, particularly the predictive insights from exogenous variables, which is vital to guarantee an accurate prediction.

## H.2    Visualization of Prediction Results

To have an intuitive concept of the forecasting process, we visually present endogenous and exogenous variables from selected datasets **BE**, **DE**, and **PJM** in Figure 12, Figure 13, and Figure

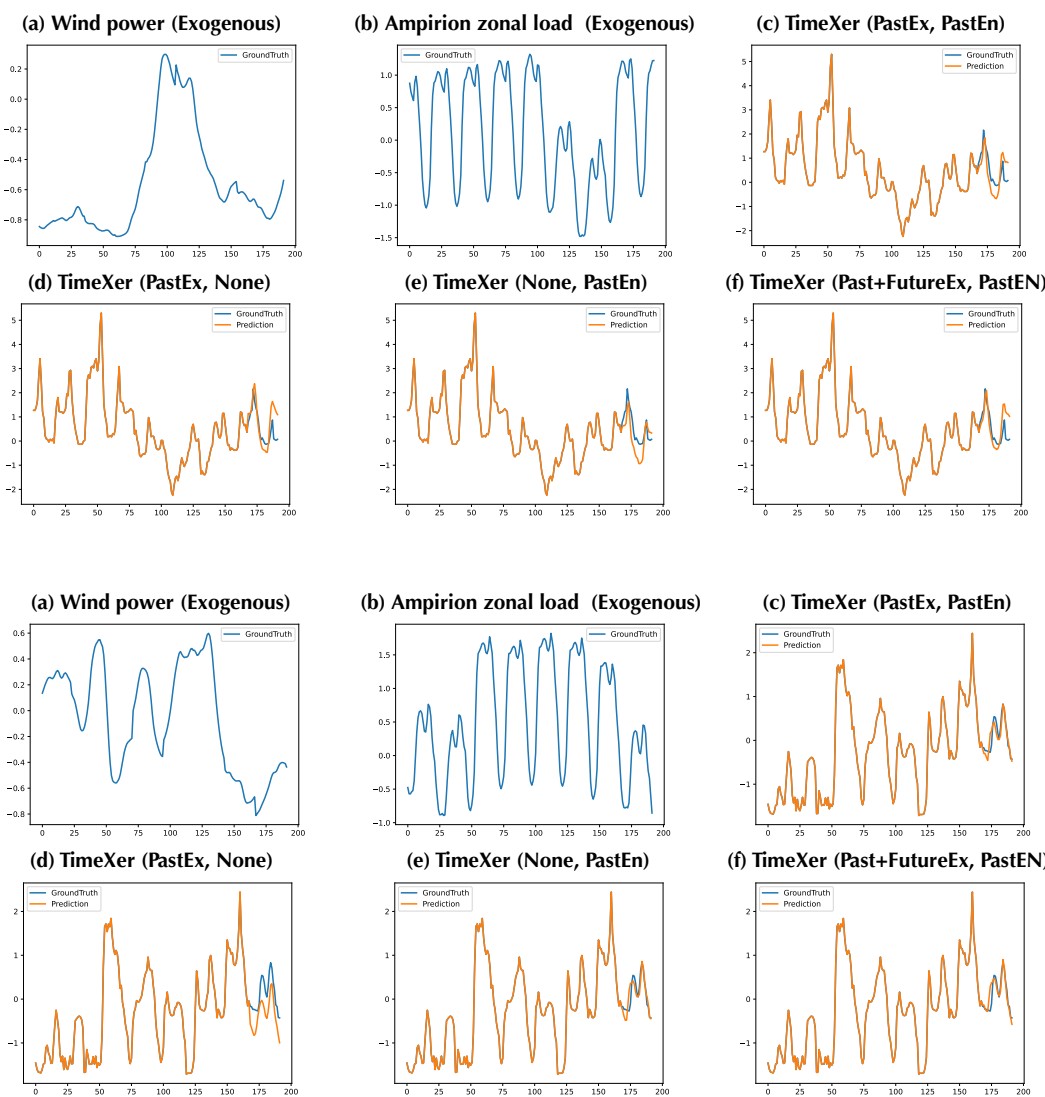

Figure 11: Two showcases (placed at the top and bottom respectively) of TimeXer in forecasting with exogenous variables from DE datasets. (a) and (b): The showcases of two exogenous variables Wind power and Ampirion zonal load. (c) The prediction results of TimeXer using the historical information of endogenous and exogenous variables. (d) and (e): The prediction results of TimeXer only use the historical information of exogenous or endogenous variables. (f) The prediction results of TimeXer using the historical information of endogenous and exogenous variables, and the future prediction values of exogenous variables.

14, respectively. For each case, we display the ground truth values for both the endogenous and exogenous variables, alongside the forecasting results for the endogenous variable. We compare the forecasting performance of the proposed TimeXer with five comparable baseline models, including Crossformer, iTransformer, PatchTST, TiDE, and DLinear. Each model takes the input time series data with a length of 168 and performs forecasting tasks with a prediction horizon of 24. To evaluate the quality of the forecasts, we utilize points of inflection on the curves. If the predicted value falls within a range of 0.05 from the ground truth, we consider this prediction successful and highlight it with a green circle of 0.05 radius. Conversely, if the predicted value exceeds this range, we classify it as an out-of-range forecast and mark it with a red circle of the same radius to indicate its failure. This visual representation facilitates a clear comparison of the performance across different models.

By counting the green and red circles on all injection points in Figure 12, Figure 13, and Figure 14, it is clear that the TimeXer can forecast target endogenous more precisely, especially at inflection points

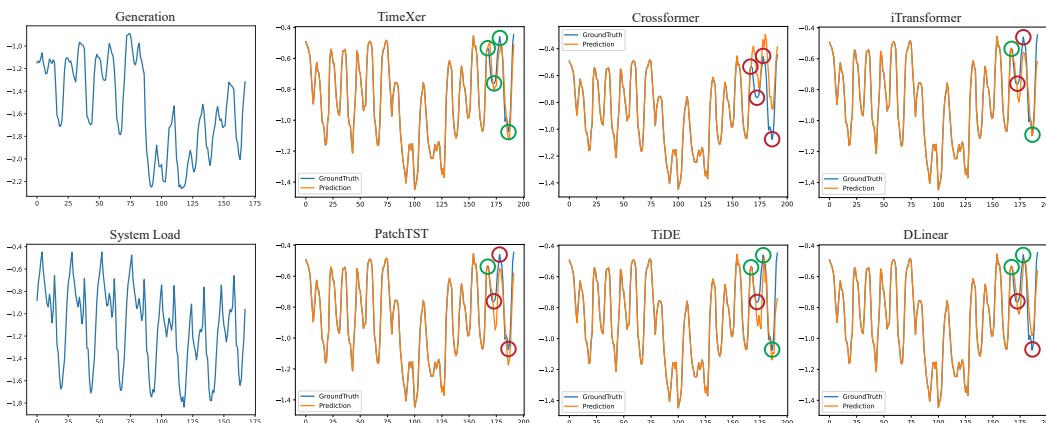

Figure 12: Showcases of TimeXer in forecasting with exogenous variables from **BE** datasets. The two leftmost plots of the title "Generation" and "System Load" are the exogenous variables in the **BE** dataset. TimeXer outperforms all of its challengers by predicting all 4 injections in 24 prediction time points.

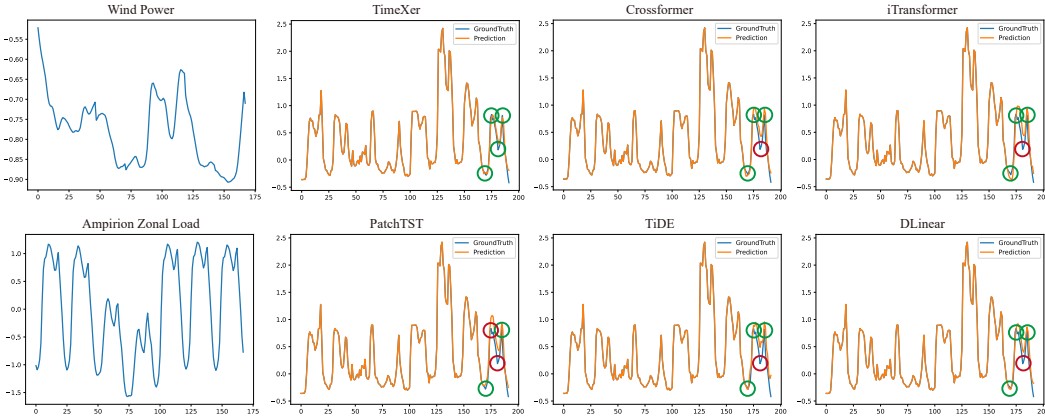

Figure 13: Showcases of TimeXer in forecasting with exogenous variables from **DE** datasets. The two leftmost plots of the title "System Load" and "Zonal COMED Load" are the exogenous variables in the **DE** dataset. TimeXer outperforms all of its challengers by predicting all 4 injections in 24 prediction time points.

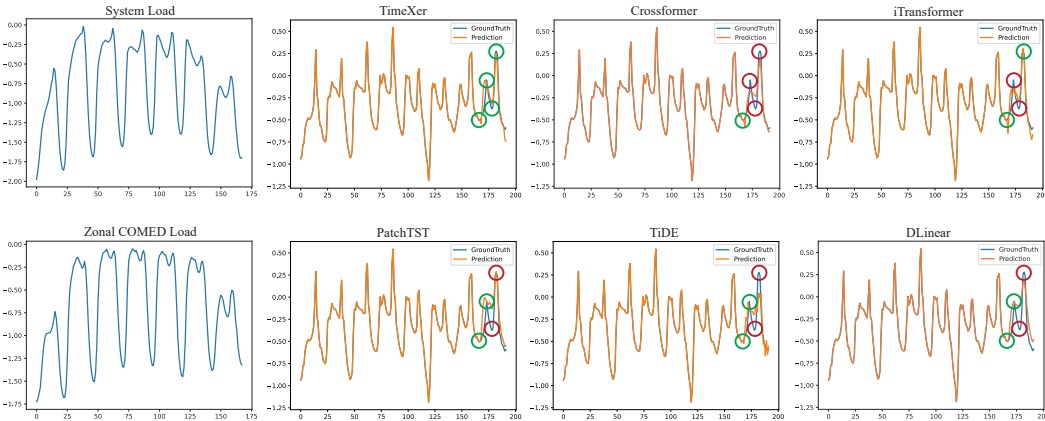

Figure 14: Showcases of TimeXer in forecasting with exogenous variables from **PJM** datasets. The two leftmost plots of the title "System Load" and "Zonal COMED Load" are the exogenous variables in the **PJM** dataset. TimeXer outperforms all of its challengers by predicting all 4 injections in 24 prediction time points.

where we marked with green circles. In contrast, other models tend to oscillate around or exceed the ground truth, suggesting that TimeXer exhibits superior robustness compared to existing alternatives. By learning from the endogenous variable's temporal dependencies and relations between endogenous and exogenous variables, TimeXer not only acquires abundant contextual information about its own history but also obtains nutritive relation information about correlated variables. Such architectural design makes TimeXer more aware of the potential pattern of the target dataset, leading to enhanced forecasting performance compared to known Transformer-based models.

# I  Full Results

## I.1  More Baselines in Short-term Forecasting

To better evaluate the performance of our proposed TimeXer, we take previous works designed for the inclusion of exogenous variables as our baselines. We also report the standard deviation of TimeXer performance on EPF dataset under five runs with different random seeds. The result in Table 9 indicates that the performance of TimeXer is stable.

Table 9: More baselines in the short-term forecasting with exogenous variables task.

| Model | NP | | PJM | | BE | | FR | | DE | | AVG | |
|---|---|---|---|---|---|---|---|---|---|---|---|---|
| Metric | MSE | MAE | MSE | MAE | MSE | MAE | MSE | MAE | MSE | MAE | MSE | MAE |
| TimeXer | 0.236±0.004 | 0.268±0.002 | 0.093±0.003 | 0.192±0.003 | 0.379±0.003 | 0.243±0.001 | 0.385±0.005 | 0.208±0.001 | 0.440±0.003 | 0.415±0.002 | 0.307±0.002 | 0.265±0.001 |
| Stationary [24] | 0.294 | 0.308 | 0.122 | 0.228 | 0.433 | 0.289 | 0.466 | 0.242 | 0.483 | 0.447 | 0.360 | 0.303 |
| NBEATSx [29] | 0.272 | 0.301 | 0.097 | 0.189 | 0.389 | 0.265 | 0.393 | 0.211 | 0.499 | 0.447 | 0.330 | 0.283 |
| TFT [20] | 0.369 | 0.391 | 0.141 | 0.241 | 0.479 | 0.305 | 0.461 | 0.249 | 0.559 | 0.490 | 0.402 | 0.335 |

## I.2  More Baselines in Long-term Forecasting

Beyond Transformer-based architecture, GNN-based models have emerged as a potential choice for modeling the underlying dynamic spatial correlations between time series. To fully evaluate the performance of TimeXer, we include classic GNN-based models, including MTGNN [39], CrossGNN [12], MSGNet [3], and FourierGNN [40] as our baselines for multivariate forecasting. The result in Table 10 shows that TimeXer consistently achieves the best.

Table 10: More baselines in the multivariate long-term forecasting. '-' denotes there is no officially reported results.

| Model | ECL | | Weather | | ETTh1 | | ETTh2 | | ETTm1 | | ETTm2 | | Traffic | |
|---|---|---|---|---|---|---|---|---|---|---|---|---|---|---|
| Metric | MSE | MAE | MSE | MAE | MSE | MAE | MSE | MAE | MSE | MAE | MSE | MAE | MSE | MAE |
| TimeXer | 0.171 | 0.270 | 0.241 | 0.271 | 0.437 | 0.437 | 0.366 | 0.395 | 0.382 | 0.397 | 0.274 | 0.322 | 0.467 | 0.288 |
| MTGNN [39] | 0.251 | 0.347 | 0.314 | 0.355 | 0.572 | 0.553 | 0.465 | 0.509 | 0.468 | 0.446 | 0.324 | 0.365 | 0.650 | 0.446 |
| CrossGNN [12] | 0.201 | 0.271 | 0.247 | 0.289 | 0.437 | 0.434 | 0.363 | 0.418 | 0.393 | 0.404 | 0.282 | 0.330 | 0.583 | 0.323 |
| MSGNet [3] | 0.194 | 0.300 | 0.249 | 0.278 | 0.0.452 | 0.452 | 0.396 | 0.417 | 0.398 | 0.411 | 0.288 | 0.330 | - | - |
| FourierGNN [40] | 0.228 | 0.324 | 0.249 | 0.302 | - | - | - | - | - | - | - | - | 0.557 | 0.342 |

## I.3  Full Results of Long-term Forecasting with exogenous variables

To evaluate the performance of our proposed TimeXer, we conduct long-term forecasting with exogenous variables on acknowledged real-world multivariate datasets. The look-back length is set to 96, and the prediction length varies from $\{96, 192, 336, 720\}$. The results are listed in Table 11.

## I.4  Full Results of Long-term Multivariate Forecasting

To evaluate the generality of our proposed TimeXer, we conduct long-term multivariate forecasting on existing real-world multivariate benchmarks. The look-back length is set to 96, and the prediction length varies from $\{96, 192, 336, 720\}$. The results are listed in Table 12.

Table 11: Full results of the long-term forecasting with exogenous variables task. "-" denotes out of memory (OOM) problem.

| Models | | TimeXer | | iTrans. | | RLinear | | PatchTST | | Cross. | | TiDE | | TimesNet | | DLinear | | SCINet | | Stationary | | Auto. | |
|---|---|---|---|---|---|---|---|---|---|---|---|---|---|---|---|---|---|---|---|---|---|---|---|
| | Metric | MSE | MAE | MSE | MAE | MSE | MAE | MSE | MAE | MSE | MAE | MSE | MAE | MSE | MAE | MSE | MAE | MSE | MAE | MSE | MAE | MSE | MAE |
| ECL | 96 | **0.261** | 0.366 | 0.299 | 0.403 | 0.433 | 0.480 | 0.339 | 0.412 | 0.265 | **0.364** | 0.405 | 0.459 | 0.342 | 0.437 | 0.387 | 0.451 | 0.390 | 0.462 | 0.298 | 0.407 | 0.432 | 0.502 |
| | 192 | 0.316 | 0.397 | 0.321 | 0.413 | 0.407 | 0.461 | 0.361 | 0.425 | **0.313** | **0.390** | 0.383 | 0.442 | 0.384 | 0.461 | 0.365 | 0.436 | 0.375 | 0.456 | 0.340 | 0.433 | 0.492 | 0.492 |
| | 336 | **0.367** | **0.429** | 0.379 | 0.446 | 0.440 | 0.481 | 0.393 | 0.440 | 0.380 | 0.431 | 0.418 | 0.464 | 0.439 | 0.493 | 0.391 | 0.453 | 0.468 | 0.519 | 0.405 | 0.471 | 0.508 | 0.548 |
| | 720 | **0.365** | **0.439** | 0.461 | 0.504 | 0.495 | 0.523 | 0.482 | 0.507 | 0.418 | 0.463 | 0.471 | 0.507 | 0.473 | 0.514 | 0.428 | 0.487 | 0.477 | 0.524 | 0.444 | 0.489 | 0.547 | 0.569 |
| | AVG | **0.327** | **0.408** | 0.365 | 0.442 | 0.444 | 0.486 | 0.394 | 0.446 | 0.344 | 0.412 | 0.419 | 0.468 | 0.410 | 0.476 | 0.393 | 0.457 | 0.427 | 0.490 | 0.372 | 0.450 | 0.495 | 0.528 |
| Weather | 96 | 0.001 | 0.027 | 0.001 | 0.026 | **0.001** | **0.025** | 0.001 | 0.027 | 0.004 | 0.048 | **0.001** | **0.025** | 0.002 | 0.029 | 0.006 | 0.062 | 0.006 | 0.064 | 0.001 | 0.028 | 0.007 | 0.066 |
| | 192 | 0.002 | 0.030 | 0.002 | 0.029 | **0.001** | **0.028** | 0.002 | 0.030 | 0.005 | 0.053 | **0.001** | **0.028** | 0.002 | 0.031 | 0.006 | 0.066 | 0.007 | 0.071 | 0.002 | 0.030 | 0.007 | 0.061 |
| | 336 | 0.002 | 0.031 | 0.002 | 0.031 | **0.002** | **0.029** | 0.002 | 0.032 | 0.004 | 0.051 | **0.002** | **0.029** | 0.002 | 0.031 | 0.006 | 0.068 | 0.008 | 0.072 | 0.002 | 0.030 | 0.007 | 0.062 |
| | 720 | 0.002 | 0.036 | 0.002 | 0.036 | **0.002** | **0.033** | 0.002 | 0.036 | 0.007 | 0.067 | **0.002** | **0.033** | 0.381 | 0.368 | 0.007 | 0.070 | 0.008 | 0.074 | **0.002** | **0.033** | 0.005 | 0.053 |
| | AVG | 0.002 | 0.031 | 0.002 | 0.031 | **0.002** | **0.029** | 0.002 | 0.031 | 0.005 | 0.055 | **0.002** | **0.029** | 0.097 | 0.115 | 0.006 | 0.066 | 0.007 | 0.071 | 0.002 | 0.031 | 0.006 | 0.060 |
| ETTh1 | 96 | 0.057 | 0.181 | 0.057 | 0.183 | 0.059 | 0.185 | **0.055** | **0.178** | 0.133 | 0.297 | 0.059 | 0.184 | 0.059 | 0.188 | 0.065 | 0.188 | 0.343 | 0.502 | 0.072 | 0.204 | 0.119 | 0.263 |
| | 192 | **0.071** | **0.204** | 0.074 | 0.209 | 0.078 | 0.214 | 0.072 | 0.206 | 0.232 | 0.409 | 0.078 | 0.214 | 0.080 | 0.217 | 0.088 | 0.222 | 0.393 | 0.533 | 0.095 | 0.238 | 0.132 | 0.286 |
| | 336 | **0.080** | **0.223** | 0.084 | 0.223 | 0.093 | 0.240 | 0.087 | 0.231 | 0.244 | 0.423 | 0.093 | 0.240 | 0.083 | 0.224 | 0.110 | 0.257 | 0.406 | 0.537 | 0.110 | 0.261 | 0.126 | 0.278 |
| | 720 | **0.084** | **0.229** | 0.084 | 0.229 | 0.106 | 0.256 | 0.098 | 0.247 | 0.530 | 0.660 | 0.104 | 0.255 | 0.083 | 0.231 | 0.202 | 0.371 | 0.604 | 0.690 | 0.164 | 0.321 | 0.143 | 0.299 |
| | AVG | **0.073** | **0.209** | 0.075 | 0.211 | 0.084 | 0.224 | 0.078 | 0.215 | 0.285 | 0.447 | 0.083 | 0.223 | 0.076 | 0.215 | 0.116 | 0.259 | 0.437 | 0.565 | 0.110 | 0.256 | 0.130 | 0.282 |
| ETTh2 | 96 | **0.132** | **0.280** | 0.137 | 0.287 | 0.136 | 0.286 | 0.136 | 0.285 | 0.261 | 0.413 | 0.136 | 0.285 | 0.159 | 0.310 | 0.135 | 0.282 | 0.763 | 0.767 | 0.186 | 0.333 | 0.184 | 0.335 |
| | 192 | **0.181** | **0.333** | 0.187 | 0.341 | 0.187 | 0.339 | 0.185 | 0.337 | 1.240 | 1.028 | 0.187 | 0.339 | 0.196 | 0.351 | 0.188 | 0.335 | 1.080 | 0.929 | 0.226 | 0.375 | 0.214 | 0.364 |
| | 336 | 0.223 | 0.377 | 0.221 | 0.376 | 0.231 | 0.384 | **0.217** | **0.373** | 0.974 | 0.874 | 0.231 | 0.384 | 0.232 | 0.385 | 0.238 | 0.385 | 1.159 | 0.960 | 0.302 | 0.443 | 0.269 | 0.405 |
| | 720 | **0.220** | **0.376** | 0.253 | 0.403 | 0.267 | 0.417 | 0.229 | 0.384 | 1.633 | 1.177 | 0.267 | 0.417 | 0.254 | 0.403 | 0.336 | 0.475 | 1.615 | 1.163 | 0.335 | 0.471 | 0.303 | 0.440 |
| | AVG | **0.189** | **0.342** | 0.199 | 0.352 | 0.205 | 0.356 | 0.192 | 0.345 | 1.027 | 0.873 | 0.205 | 0.356 | 0.210 | 0.362 | 0.224 | 0.369 | 1.155 | 0.955 | 0.262 | 0.405 | 0.242 | 0.386 |
| ETTm1 | 96 | **0.028** | **0.125** | 0.029 | 0.128 | 0.030 | 0.129 | 0.029 | 0.126 | 0.171 | 0.355 | 0.030 | 0.129 | 0.029 | 0.128 | 0.034 | 0.135 | 0.050 | 0.173 | 0.034 | 0.138 | 0.097 | 0.251 |
| | 192 | **0.043** | **0.158** | 0.045 | 0.163 | 0.044 | 0.160 | 0.045 | 0.160 | 0.293 | 0.474 | 0.044 | 0.160 | 0.044 | 0.160 | 0.055 | 0.173 | 0.083 | 0.227 | 0.060 | 0.182 | 0.062 | 0.197 |
| | 336 | 0.058 | 0.185 | 0.060 | 0.190 | **0.057** | 0.184 | 0.058 | **0.184** | 0.330 | 0.503 | **0.057** | **0.184** | 0.061 | 0.190 | 0.078 | 0.210 | 0.110 | 0.261 | 0.087 | 0.222 | 0.083 | 0.230 |
| | 720 | **0.079** | **0.217** | **0.079** | 0.218 | 0.080 | **0.217** | 0.082 | 0.221 | 0.852 | 0.861 | 0.080 | **0.217** | 0.083 | 0.223 | 0.098 | 0.234 | 0.152 | 0.305 | 0.127 | 0.275 | 0.100 | 0.245 |
| | AVG | **0.052** | **0.171** | 0.053 | 0.175 | 0.053 | 0.173 | 0.053 | 0.173 | 0.411 | 0.548 | 0.053 | 0.173 | 0.054 | 0.175 | 0.066 | 0.188 | 0.098 | 0.241 | 0.077 | 0.204 | 0.085 | 0.230 |
| ETTm2 | 96 | **0.067** | **0.188** | 0.071 | 0.194 | 0.074 | 0.199 | 0.068 | **0.188** | 0.149 | 0.309 | 0.073 | 0.199 | 0.073 | 0.200 | 0.072 | 0.195 | 0.253 | 0.427 | 0.098 | 0.229 | 0.133 | 0.282 |
| | 192 | 0.101 | **0.236** | 0.108 | 0.247 | 0.104 | 0.241 | **0.100** | **0.236** | 0.686 | 0.740 | 0.104 | 0.241 | 0.106 | 0.247 | 0.105 | 0.240 | 0.592 | 0.677 | 0.161 | 0.302 | 0.143 | 0.294 |
| | 336 | 0.130 | 0.275 | 0.140 | 0.288 | 0.131 | 0.276 | **0.128** | **0.271** | 0.546 | 0.602 | 0.131 | 0.276 | 0.150 | 0.296 | 0.136 | 0.280 | 0.777 | 0.790 | 0.243 | 0.362 | 0.156 | 0.308 |
| | 720 | 0.182 | 0.332 | 0.188 | 0.340 | **0.180** | **0.329** | 0.185 | 0.335 | 2.524 | 1.424 | **0.180** | **0.329** | 0.186 | 0.338 | 0.191 | 0.335 | 1.117 | 0.960 | 0.326 | 0.441 | 0.184 | 0.333 |
| | AVG | **0.120** | **0.258** | 0.127 | 0.267 | 0.122 | 0.261 | **0.120** | **0.258** | 0.976 | 0.769 | 0.122 | 0.261 | 0.129 | 0.271 | 0.126 | 0.263 | 0.685 | 0.713 | 0.207 | 0.333 | 0.154 | 0.305 |
| Traffic | 96 | **0.151** | **0.224** | 0.156 | 0.236 | 0.350 | 0.431 | 0.176 | 0.253 | 0.154 | 0.230 | 0.350 | 0.430 | 0.154 | 0.249 | 0.268 | 0.351 | 0.371 | 0.448 | 0.214 | 0.323 | 0.290 | 0.290 |
| | 192 | **0.152** | **0.229** | 0.156 | 0.237 | 0.314 | 0.404 | 0.162 | 0.243 | 0.180 | 0.256 | 0.230 | 0.315 | 0.164 | 0.255 | 0.302 | 0.387 | 0.450 | 0.503 | 0.195 | 0.307 | 0.291 | 0.291 |
| | 336 | **0.150** | **0.232** | 0.154 | 0.243 | 0.305 | 0.399 | 0.164 | 0.248 | - | - | 0.220 | 0.208 | 0.167 | 0.259 | 0.298 | 0.384 | 0.447 | 0.501 | 0.198 | 0.309 | 0.322 | 0.416 |
| | 720 | **0.172** | **0.253** | 0.177 | 0.268 | 0.328 | 0.415 | 0.189 | 0.267 | - | - | 0.243 | 0.329 | 0.197 | 0.292 | 0.340 | 0.416 | 0.521 | 0.548 | 0.835 | 0.507 | 0.307 | 0.414 |
| | AVG | **0.156** | **0.234** | 0.161 | 0.246 | 0.324 | 0.412 | 0.173 | 0.253 | - | - | 0.240 | 0.326 | 0.171 | 0.264 | 0.323 | 0.404 | 0.447 | 0.500 | 0.361 | 0.361 | 0.302 | 0.353 |
| 1st Count | | 23 | 23 | 2 | 1 | 7 | 8 | 5 | 5 | 1 | 2 | 7 | 8 | 0 | 0 | 0 | 0 | 0 | 0 | 1 | 1 | 0 | 0 |

Table 12: Full results of the long-term multivariate forecasting task.

| Models | Metric | TimeXer MSE | TimeXer MAE | iTrans. MSE | iTrans. MAE | RLinear MSE | RLinear MAE | PatchTST MSE | PatchTST MAE | Cross. MSE | Cross. MAE | TiDE MSE | TiDE MAE | TimesNet MSE | TimesNet MAE | DLinear MSE | DLinear MAE | SCINet MSE | SCINet MAE | Stationary MSE | Stationary MAE | Auto. MSE | Auto. MAE |
|---|---|---|---|---|---|---|---|---|---|---|---|---|---|---|---|---|---|---|---|---|---|---|---|
| ECL | 96 | **0.140** | 0.242 | 0.148 | **0.240** | 0.201 | 0.281 | 0.195 | 0.285 | 0.219 | 0.314 | 0.237 | 0.329 | 0.168 | 0.272 | 0.197 | 0.282 | 0.247 | 0.345 | 0.169 | 0.273 | 0.201 | 0.317 |
| | 192 | **0.157** | 0.256 | 0.162 | **0.253** | 0.201 | 0.283 | 0.199 | 0.289 | 0.231 | 0.322 | 0.236 | 0.330 | 0.184 | 0.289 | 0.196 | 0.285 | 0.257 | 0.355 | 0.182 | 0.286 | 0.222 | 0.334 |
| | 336 | **0.176** | 0.275 | 0.178 | **0.269** | 0.215 | 0.298 | 0.215 | 0.305 | 0.246 | 0.337 | 0.249 | 0.344 | 0.198 | 0.300 | 0.209 | 0.301 | 0.269 | 0.369 | 0.200 | 0.304 | 0.231 | 0.338 |
| | 720 | **0.211** | **0.306** | 0.225 | 0.317 | 0.257 | 0.331 | 0.256 | 0.337 | 0.280 | 0.363 | 0.284 | 0.373 | 0.220 | 0.320 | 0.245 | 0.333 | 0.299 | 0.390 | 0.222 | 0.321 | 0.254 | 0.361 |
| | Avg | **0.171** | **0.270** | 0.178 | **0.270** | 0.219 | 0.298 | 0.216 | 0.304 | 0.244 | 0.334 | 0.251 | 0.344 | 0.192 | 0.295 | 0.212 | 0.300 | 0.268 | 0.365 | 0.193 | 0.296 | 0.227 | 0.338 |
| Weather | 96 | **0.157** | **0.205** | 0.174 | 0.214 | 0.192 | 0.232 | 0.177 | 0.218 | 0.158 | 0.230 | 0.202 | 0.261 | 0.172 | 0.220 | 0.196 | 0.255 | 0.221 | 0.306 | 0.173 | 0.223 | 0.266 | 0.336 |
| | 192 | **0.204** | **0.247** | 0.221 | 0.254 | 0.240 | 0.271 | 0.225 | 0.259 | 0.206 | 0.277 | 0.242 | 0.298 | 0.219 | 0.261 | 0.237 | 0.296 | 0.261 | 0.340 | 0.245 | 0.285 | 0.307 | 0.367 |
| | 336 | **0.261** | **0.290** | 0.278 | 0.296 | 0.292 | 0.307 | 0.278 | 0.297 | 0.272 | 0.335 | 0.287 | 0.335 | 0.280 | 0.306 | 0.283 | 0.335 | 0.309 | 0.378 | 0.321 | 0.338 | 0.359 | 0.395 |
| | 720 | **0.340** | **0.341** | 0.358 | 0.349 | 0.364 | 0.353 | 0.354 | 0.348 | 0.398 | 0.418 | 0.351 | 0.386 | 0.365 | 0.359 | 0.345 | 0.381 | 0.377 | 0.427 | 0.414 | 0.410 | 0.419 | 0.428 |
| | Avg | **0.241** | **0.271** | 0.258 | 0.279 | 0.272 | 0.291 | 0.259 | 0.281 | 0.259 | 0.315 | 0.271 | 0.320 | 0.259 | 0.287 | 0.265 | 0.317 | 0.292 | 0.363 | 0.288 | 0.314 | 0.338 | 0.382 |
| ETTh1 | 96 | **0.382** | 0.403 | 0.386 | 0.405 | 0.386 | **0.395** | 0.414 | 0.419 | 0.423 | 0.448 | 0.479 | 0.464 | 0.384 | 0.402 | 0.386 | 0.400 | 0.654 | 0.599 | 0.513 | 0.491 | 0.449 | 0.459 |
| | 192 | **0.429** | 0.435 | 0.441 | 0.436 | 0.437 | **0.424** | 0.460 | 0.445 | 0.471 | 0.474 | 0.525 | 0.492 | 0.436 | 0.429 | 0.437 | 0.432 | 0.719 | 0.631 | 0.534 | 0.504 | 0.500 | 0.482 |
| | 336 | **0.468** | 0.448 | 0.487 | 0.458 | 0.479 | **0.446** | 0.501 | 0.466 | 0.570 | 0.546 | 0.565 | 0.515 | 0.491 | 0.469 | 0.481 | 0.459 | 0.778 | 0.659 | 0.588 | 0.535 | 0.521 | 0.496 |
| | 720 | **0.469** | **0.461** | 0.503 | 0.491 | 0.481 | 0.470 | 0.500 | 0.488 | 0.653 | 0.621 | 0.594 | 0.558 | 0.521 | 0.500 | 0.519 | 0.516 | 0.836 | 0.699 | 0.643 | 0.616 | 0.514 | 0.512 |
| | Avg | **0.437** | 0.437 | 0.454 | 0.447 | 0.446 | **0.434** | 0.469 | 0.454 | 0.529 | 0.522 | 0.541 | 0.507 | 0.458 | 0.450 | 0.456 | 0.452 | 0.747 | 0.647 | 0.570 | 0.537 | 0.496 | 0.487 |
| ETTh2 | 96 | **0.286** | 0.338 | 0.297 | 0.349 | 0.288 | **0.338** | 0.302 | 0.348 | 0.745 | 0.584 | 0.400 | 0.440 | 0.340 | 0.374 | 0.333 | 0.387 | 0.707 | 0.621 | 0.476 | 0.458 | 0.346 | 0.388 |
| | 192 | **0.363** | **0.389** | 0.380 | 0.400 | 0.374 | 0.390 | 0.388 | 0.400 | 0.877 | 0.656 | 0.528 | 0.509 | 0.402 | 0.414 | 0.477 | 0.476 | 0.860 | 0.689 | 0.512 | 0.493 | 0.456 | 0.452 |
| | 336 | **0.414** | **0.423** | 0.428 | 0.432 | 0.415 | 0.426 | 0.426 | 0.433 | 1.043 | 0.731 | 0.643 | 0.571 | 0.452 | 0.452 | 0.594 | 0.541 | 1.000 | 0.744 | 0.552 | 0.551 | 0.482 | 0.486 |
| | 720 | **0.408** | **0.432** | 0.427 | 0.445 | 0.420 | 0.440 | 0.431 | 0.446 | 1.104 | 0.763 | 0.874 | 0.679 | 0.462 | 0.468 | 0.831 | 0.657 | 1.249 | 0.838 | 0.562 | 0.560 | 0.515 | 0.511 |
| | Avg | **0.367** | **0.396** | 0.383 | 0.407 | 0.374 | 0.398 | 0.387 | 0.407 | 0.942 | 0.684 | 0.611 | 0.550 | 0.414 | 0.427 | 0.559 | 0.515 | 0.954 | 0.723 | 0.526 | 0.516 | 0.450 | 0.459 |
| ETTm1 | 96 | **0.318** | **0.356** | 0.334 | 0.368 | 0.355 | 0.376 | 0.329 | 0.367 | 0.404 | 0.426 | 0.364 | 0.387 | 0.338 | 0.375 | 0.345 | 0.372 | 0.418 | 0.438 | 0.386 | 0.398 | 0.505 | 0.475 |
| | 192 | **0.362** | **0.383** | 0.387 | 0.391 | 0.391 | 0.392 | 0.367 | 0.385 | 0.450 | 0.451 | 0.398 | 0.404 | 0.374 | 0.387 | 0.380 | 0.389 | 0.426 | 0.441 | 0.459 | 0.444 | 0.553 | 0.496 |
| | 336 | **0.395** | **0.407** | 0.426 | 0.420 | 0.424 | 0.415 | 0.399 | 0.410 | 0.532 | 0.515 | 0.428 | 0.425 | 0.410 | 0.411 | 0.413 | 0.413 | 0.445 | 0.459 | 0.495 | 0.464 | 0.621 | 0.537 |
| | 720 | **0.452** | 0.441 | 0.491 | 0.459 | 0.487 | 0.450 | 0.454 | **0.439** | 0.666 | 0.589 | 0.487 | 0.461 | 0.478 | 0.450 | 0.474 | 0.453 | 0.595 | 0.550 | 0.585 | 0.516 | 0.671 | 0.561 |
| | Avg | **0.382** | **0.397** | 0.407 | 0.410 | 0.414 | 0.407 | 0.387 | 0.400 | 0.513 | 0.496 | 0.419 | 0.419 | 0.400 | 0.406 | 0.403 | 0.407 | 0.485 | 0.481 | 0.481 | 0.456 | 0.588 | 0.517 |
| ETTm2 | 96 | **0.171** | **0.256** | 0.180 | 0.264 | 0.182 | 0.265 | 0.175 | 0.259 | 0.287 | 0.366 | 0.207 | 0.305 | 0.187 | 0.267 | 0.193 | 0.292 | 0.286 | 0.377 | 0.192 | 0.274 | 0.255 | 0.339 |
| | 192 | **0.237** | **0.299** | 0.250 | 0.309 | 0.246 | 0.304 | 0.241 | 0.302 | 0.414 | 0.492 | 0.290 | 0.364 | 0.249 | 0.309 | 0.284 | 0.362 | 0.399 | 0.445 | 0.280 | 0.339 | 0.281 | 0.340 |
| | 336 | **0.296** | **0.338** | 0.311 | 0.348 | 0.307 | 0.342 | 0.305 | 0.343 | 0.597 | 0.542 | 0.377 | 0.422 | 0.321 | 0.351 | 0.369 | 0.427 | 0.637 | 0.591 | 0.334 | 0.361 | 0.339 | 0.372 |
| | 720 | **0.392** | **0.394** | 0.412 | 0.407 | 0.407 | 0.398 | 0.402 | 0.400 | 1.730 | 1.042 | 0.558 | 0.524 | 0.408 | 0.403 | 0.554 | 0.522 | 0.960 | 0.735 | 0.417 | 0.413 | 0.433 | 0.432 |
| | Avg | **0.274** | **0.322** | 0.288 | 0.332 | 0.286 | 0.327 | 0.281 | 0.326 | 0.757 | 0.610 | 0.358 | 0.404 | 0.291 | 0.333 | 0.350 | 0.401 | 0.571 | 0.537 | 0.306 | 0.347 | 0.327 | 0.371 |
| Traffic | 96 | 0.428 | 0.271 | **0.395** | **0.268** | 0.649 | 0.389 | 0.462 | 0.295 | 0.522 | 0.290 | 0.805 | 0.493 | 0.593 | 0.321 | 0.650 | 0.396 | 0.788 | 0.499 | 0.612 | 0.338 | 0.613 | 0.388 |
| | 192 | 0.448 | 0.282 | **0.417** | **0.276** | 0.601 | 0.366 | 0.466 | 0.296 | 0.530 | 0.293 | 0.756 | 0.474 | 0.617 | 0.336 | 0.598 | 0.370 | 0.789 | 0.505 | 0.613 | 0.340 | 0.616 | 0.382 |
| | 336 | 0.473 | 0.289 | **0.433** | **0.283** | 0.609 | 0.369 | 0.482 | 0.304 | 0.558 | 0.305 | 0.762 | 0.477 | 0.629 | 0.336 | 0.605 | 0.373 | 0.797 | 0.508 | 0.618 | 0.328 | 0.622 | 0.337 |
| | 720 | 0.516 | 0.307 | **0.467** | **0.302** | 0.647 | 0.387 | 0.514 | 0.322 | 0.589 | 0.328 | 0.719 | 0.449 | 0.640 | 0.350 | 0.645 | 0.394 | 0.841 | 0.523 | 0.653 | 0.355 | 0.660 | 0.408 |
| | Avg | 0.466 | 0.287 | **0.428** | **0.282** | 0.626 | 0.378 | 0.481 | 0.304 | 0.550 | 0.304 | 0.760 | 0.473 | 0.620 | 0.336 | 0.625 | 0.383 | 0.804 | 0.509 | 0.624 | 0.340 | 0.628 | 0.379 |
| 1st Count | | 30 | 22 | 5 | 9 | 0 | 5 | 0 | 1 | 0 | 0 | 0 | 0 | 0 | 0 | 0 | 0 | 0 | 0 | 0 | 0 | 0 | 0 |

