# OpenReview forum: "TimeXer: Empowering Transformers for Time Series Forecasting with Exogenous Variables"
_NeurIPS.cc/2024/Conference — NeurIPS 2024 poster_

### Official Review · Reviewer_N2A5 · 2024-06-29

**Soundness:** 3
**Presentation:** 4
**Contribution:** 3
**Rating:** 8
**Confidence:** 4

**Summary:**

The paper introduces `TimeXer`, a novel approach for time series forecasting with exogenous variables. Its main contributions include a new Transformer-based model architecture designed to effectively integrate and utilize exogenous information alongside endogenous time series data. The paper demonstrates the effectiveness of `TimeXer` through extensive experiments on various real-world datasets, showcasing its superior performance compared to **state-of-the-art** methods.

**Strengths:**

- The paper introduces a novel approach to **integrating exogenous variables** into **Transformer-based** models for time series forecasting.
- The model design is well-thought-out, and the experiments are carefully executed.
- The paper is well-written and clearly articulates the problem, solution, and experimental results.
- The work has potential implications for improving forecasting accuracy in various real-world applications.

**Weaknesses:**

- There is no comparison between native models that can be used for forecasting with Exogenous Variables.

**Questions:**

- How is the memory consumption of the model measured? It seems that there is no implementation in the provided code.
- The prediction task of TimeXer and previous prediction task such as `iTransformer` and `DLinear` are essentially different types of prediction targets. More details need to be disclosed on how to transfer previous prediction tasks in the text.

**Limitations:**

- The paper adequately discusses the limitations, such as the need for more extensive experimentation and the potential challenges with a large number of exogenous variables.

---

> ### Author Rebuttal · Authors · 2024-08-06
>
> # Response to Reviewer N2A5
>
> We would like to sincerely thank Reviewer N2A5 for providing a detailed review and insightful suggestions.
>
> > **W1**: There is no comparison between native models that can be used for forecasting with Exogenous Variables.
>
> In $\underline{\text{Table 7 of the Appendix}}$, we have compared the performance of TimeXer with previous approaches, NBEATSx and TFT, both of which are natively designed for forecasting with exogenous variables. To make this clearer, we present the results here and TimeXer surpasses these two native models.
>
> |Short-term(MSE\|MAE)| NP           | PJM          | BE           | FR           | DE           | AVG          |
> |-|-|-|-|-|-|-|
> | TimeXer    | **0.236** \| **0.268** | **0.093** \| **0.192** | **0.379** \| **0.243** | **0.385** \| **0.208** | **0.440** \| **0.415** | **0.307** \| **0.265** |
> | NBeatsX     | 0.272 \| 0.301 | 0.097 \| 0.189 | 0.389 \| 0.265 | 0.393 \| 0.211 | 0.499 \| 0.447 | 0.330 \| 0.283 |
> | TFT     | 0.369 \| 0.391 | 0.141 \| 0.241 | 0.479 \| 0.305 | 0.461 \| 0.249 | 0.559 \| 0.490 | 0.402 \| 0.335 |
>
>
> > **Q1**: How is the memory consumption of the model measured? It seems that there is no implementation in the provided code.
>
> We assess memory consumption by monitoring GPU memory usage during the model training phase.
>
> > **Q2**: The prediction task of TimeXer and previous prediction task such as iTransformer and DLinear are essentially different types of prediction targets. More details need to be disclosed on how to transfer previous prediction tasks in the text.
>
> Sorry for the missing details. As stated in $\underline{\text{Line 193 of main text}}$, we discover that the forecasting with exogenous variables task can be generalized into multivariate forecasting tasks through the channel independence mechanism. Technologically, we consider the variables in the dataset as mutually independent endogenous variables, with each variable considering all other variables except itself as exogenous. We also have provided the code for multivariate forecasting in the $\underline{\text{supplementary materials}}$.

---

> > ### Comment · Reviewer_N2A5 · 2024-08-13
> >
> > Thank you for the clarifications and additional details. The comparison with NBEATSx and TFT is informative and strengthens the manuscript. The explanation of memory consumption measurement and the generalization of forecasting tasks are satisfactory. The revisions address the main concerns raised. I maintain my strong recommendation for acceptance.

---

> ### Comment · Area_Chair_qCjj · 2024-08-12
> **author-reviewer discussion**
>
> Dear reviewer,
>
> The author-reviewer discussion ends soon. If you need additional clarifications from the authors, please respond to the authors asap. Thank you very much.
>
> Best,
>
> AC

---

> ### Comment · Reviewer_N2A5 · 2024-08-13
>
> I will complete my response promptly. Thank you for the reminder.

---

### Official Review · Reviewer_1zKu · 2024-07-07

**Soundness:** 3
**Presentation:** 2
**Contribution:** 2
**Rating:** 4
**Confidence:** 4

**Summary:**

The article designs TimeXer, Empowering Transformers for Time Series Forecasting with Exogenous Variables. TimeXer effectively utilizes exogenous information to enhance the accuracy of time series forecasting. Extensive experimental results validate the effectiveness of the proposed method.

**Strengths:**

S1. Unlike general multivariate time series forecasting, this paper introduces a new task for time series forecasting by integrating exogenous variables to predict endogenous sequences, which is a meaningful research direction.

S2. The experiments are comprehensive, with detailed validations on datasets from different domains and under various prediction horizons.

S3. The method is simple and intuitively effective.

S4. The scalability validation in Figure 4 is interesting and convincing.

**Weaknesses:**

W1. There is a potentially slight inconsistency between the description of the method and the scale of the data used in the experiments. The method section suggests that the length of the exogenous variables differs from that of the endogenous variables. However, in the datasets used for both long-term and short-term forecasting, the lengths of the exogenous and endogenous variables are the same.

W2. Although the authors extensively discuss related work on considering relationships between variables in time series forecasting, some highly relevant literature is missing [1,2]. It is recommended to include these in the camera-ready version.

[1] CrossGNN: Confronting Noisy Multivariate Time Series Via Cross Interaction Refinement. NeurIPS 2023.

[2] FourierGNN: Rethinking Multivariate Time Series Forecasting from a Pure Graph Perspective. NeurIPS 2023.

W3. Figure 5 presents the memory usage of TimeXer and some baselines. Essentially, TimeXer and iTransformer share a similar model structure, yet TimeXer consumes less memory. What do you believe accounts for TimeXer being more memory-efficient than iTransformer?

W4. Table 4 presents a comprehensive ablation study. However, I would like to see the complete removal of the exogenous variable enhancement module, for instance, by setting the exogenous variables to zero or random numbers, to explore TimeXer's adaptability to exogenous variables. If the performance deteriorates, it may indicate that TimeXer's results depend on the quality of the exogenous variables. If the performance remains stable, it suggests that TimeXer has good adaptability.

**Questions:**

Q1. Have you considered exploiting large language models (LLM) and leveraging rich textual knowledge, to improve the model's ability to understand exogenous variables at multiple granularities?

Q2. In most forecasting scenarios, do you think that the historical information of the target sequence itself is more useful, or is the information from exogenous variables more valuable? You can discuss on it freely.

**Limitations:**

The authors have thoroughly discussed the limitations of this work in the appendix.

---

> ### Author Rebuttal · Authors · 2024-08-06
>
> # Response to Reviewer 1zKu
>
> Many thanks to Reviewer 1zKu for providing a detailed review and questions.
> > **W1**: There is a potentially slight inconsistency between the description of the method and the scale of the data used in the experiments.
>
> Thank you for your careful reading. Since most of the baseline forecasters can only support cases where the lengths of the exogenous and endogenous variables are the same, we set the lengths of the exogenous and endogenous variables to be the same to align with previous benchmarks for a fair comparison.
>
> We have **experimented with the case where the exogenous and endogenous variables are of unequal length** in $\underline{\text{Section 4.3 of main text}}$, encompassing two common real-world scenarios: (1) unequal look-back length (2) different sampling frequencies. As presented in $\underline{\text{Figures 3,4 of main text}}$, TimeXer exhibits superior generality over various practical cases.
> > **W2**: Some highly relevant literature is missing.
>
> Thank you for the recommendation. We will include them as baselines in the camera-ready version. Considering these two models are proposed for multivariate data, we compare the TimeXer performance in multivariate forecasting with those given in their original papers. Note that the experimental setup in FourierGNN is not the same as TimeXer, we use the results reported by ForecastGrapher[1].
>
> |MSE\|MAE|Electricity|Traffic|Weather|
> |-|-|-|-|
> |TimeXer|**0.172**\|**0.272**|**0.472**\|**0.283**|**0.242**\|**0.283**|
> |CrossGNN|0.201\|0.297|0.583\|0.323|0.247\|0.289|
> |FourierGNN|0.228\|0.324|0.557\|0.342|0.249\|0.302|
>
> [1] "ForecastGrapher: Redefining Multivariate Time Series Forecasting with Graph Neural Networks." arXiv preprint arXiv:2405.18036 (2024).
> > **W3**: Essentially, TimeXer and iTransformer share a similar model structure, yet TimeXer consumes less memory. What do you believe accounts for TimeXer being more memory-efficient than iTransformer?
>
> Many thanks for your insightful question. We have included a theoretical efficiency analysis w.r.t. iTransformer in $\underline{\text{Appendix E}}$. Here is an intuitive understanding.
>
> The key design of iTransformer is embedding each variate series into one token. Afterwards, the embedding variate tokens will be fed to Transformer, which will apply a self-attention mechanism among variate tokens. Although this design can keep refining the learned variate token in multiple layers, it does cause more complexity.
>
> As for TimeXer, as presented in $\underline{\text{Figure 2(b) of main text}}$, exogenous series will be embedded to variate tokens at the beginning, which will be shared in all layers and interact with the endogenous global token by cross attention. Thus, **TimeXer omits the interaction among learned exogenous variate tokens**, which makes TimeXer more efficient than iTransformer.
> > **W4**: I would like to see the complete removal of the exogenous variable enhancement module, for instance, by setting the exogenous variables to zero or random numbers, to explore TimeXer's adaptability to exogenous variables.
>
> We have provided the experimental results on TimeXer's generality under missing exogenous values in $\underline{\text{Appendix C}}$. As per your request, we have conducted more ablation studies on the quality of the exogenous variables.
>
> Technologically, we use two kinds of exogenous variables: (1) set the series into zeros (TimeXer-zerosEX) and (2) set the series to random numbers with mean 0 and variance 1 (TimeXer-randomEX). As demonstrated in the $\underline{\text{Table 1 in the Author Rebuttal PDF}}$, the inclusion of exogenous variables leads to an improvement in model performance, while the performance decreases when the exogenous variables are meaningless noise.
> > **Q1**: Have you considered exploiting large language models (LLM) and leveraging rich textual knowledge, to improve the model's ability to understand exogenous variables at multiple granularities?
>
> Following the reviewer's suggestion, we have endeavored to leverage large language models for forecasting with exogenous. Technologically, we design a multi-grained prompt and use a T5 tokenizer to extract textual information, which is then integrated into TimeXer by adding it to the corresponding variate tokens. The prompt consists of the name and sampling frequency of each variable, for example, "This exogenous variable is load generation, with a sampling frequency of 1h. We use it to enhance the prediction of electricity price in the Nord Pool market".
>
> |Short-term(MSE\|MAE)|NP|PJM|BE|FR|DE|AVG|
> -|-|-|-|-|-|-
> |TimeXer-LLM|0.236\|0.266|0.106\|0.198|0.382\|0.245|0.401\|0.218|0.450\|0.422|0.315\|0.270|
>
> Our experiments indicate that there was no improvement in model performance, likely due to insufficient textual information. We believe that a more sophisticated design for prompts is necessary to capture exogenous variables at multiple levels of granularity, which we will address in future work.
> > **Q2**: In most forecasting scenarios, do you think that the historical information of the target sequence itself is more useful, or is the information from exogenous variables more valuable?
>
> Thanks for the reviewer's valuable suggestion. As per your request, we set the exogenous and endogenous series to zeros respectively (TimeXer-zerosEX and TimeXer-zerosEN), and the results are listed in $\underline{\text{Table 1 in the Author Rebuttal PDF}}$. We find that TimeXer benefits more from the historical information of endogenous series.
>
> Therefore, in most forecasting scenarios, the historical information of the target sequence itself is more useful. However, there are specific cases where the exogenous variable has a time-lagged influence on the non-stationary endogenous variable, in which case the exogenous series may be more useful.

---

### Official Review · Reviewer_Mh7v · 2024-07-09

**Soundness:** 3
**Presentation:** 3
**Contribution:** 3
**Rating:** 6
**Confidence:** 5

**Summary:**

This article proposes a method based on Transformer modeling to enhance the prediction accuracy of endogenous variables by incorporating exogenous variables.

**Strengths:**

1. The research topic is interesting and has strong practical value.
2. The paper is well-structured, making it easy for readers to understand.
3. The experiments are extensive, demonstrating the effectiveness of the method across various datasets and experimental settings.

**Weaknesses:**

1. The implementation details on how to extend TimeXer to multivariate time series prediction are not sufficiently detailed (lines 190 - 196).

2. The article does not provide a specific explanation for why the non-overlapping patch method is used instead of the overlapping method.

3. Figure 3 shows the prediction performance corresponding to different step lengths of exogenous/endogenous variables, but it lacks experimental results for some extreme cases, such as reducing the step length of exogenous/endogenous variables to 1.

4. It is recommended to explain more clearly in the methods and introduction sections why the global token is applied to the time series prediction task.

**Questions:**

The influence of exogenous variables on endogenous variables is multi-grained, including aspects like periodicity, trend, noise, and stability. Does the Transformer have the capability to capture such multi-grained influences, or does the author think that more promising methods (such as LLMs) could be exploited to capture these associations?

**Limitations:**

The limitation is discussed sufficiently in appendix.

---

> ### Author Rebuttal · Authors · 2024-08-06
>
> # Response to Reviewer Mh7v
> Many thanks to Reviewer Mh7v for providing a detailed review and insightful questions.
> > **W1**: The implementation details on how to extend TimeXer to multivariate time series prediction are not sufficiently detailed.
>
> Sorry for the missing description. The success of the channel independence mechanism has demonstrated that multivariate forecasting can be viewed as multiple independent univariate forecasting with a shared backbone. Inspired by this, we find that TimeXer can be generalized to multivariate forecasting by employing the channel independence mechanism.
>
> Specifically, For multivariate time series [B, T, C], we obtain endogenous embedded vector [B\*C, N, D] and exogenous embedded vector [B, C, D], where N is the number of patches and D is the hidden dimension. Note that the endogenous embedded vector is already channel-independent and further fed into the shared Endogenous Self-Attention Module to capture temporal dependencies within each variable. Further, we select the global token of each variable (which is then reshaped from [B\*C, 1, D] to [B, C, D]) to perform cross-attention with the exogenous embedded vector to capture cross-variate dependencies. We have provided the relevant code in $\underline{\text{supplementary materials}}$.
> > **W2**: The article does not provide a specific explanation for why the non-overlapping patch method is used instead of the overlapping method.
>
> Following your suggestion, we have completed the ablation study on the effects of patching in TimeXer. We adopt the patching method used in PatchTST (ICLR 2023), setting the patch length to 24, consistent with TimeXer, and the stride to 12, to generate a sequence of overlapped patches. Compared to the overlapping method, TimeXer has the lowest complexity while having the optimal performance.
> |Short-term(MSE\|MAE)|NP|PJM|BE|FR|DE|AVG|
> -|-|-|-|-|-|-
> |TimeXer|**0.236**\|**0.268**|**0.093**\|**0.192**|**0.379**\|**0.243**|**0.385**\|**0.208**|**0.440**\|**0.415**|**0.307**\|**0.265**
> |TimeXer-overlap|0.240\|0.267|0.095\|0.194|0.383\|0.248|0.409\|0.214|0.453\|0.419|0.316\|0.269|
>
> It is also notable that not only in NLP and CV, contemporary time series approaches ([41 of our paper] and [1]) also use non-overlapping patches. This preference might stem from the limited redundancy present in time series data, as excessive overlap can result in a smoothed representation for each patch, consequently failing to capture correct temporal dependencies.
>
> [1] "Timer: Generative Pre-trained Transformers Are Large Time Series Models." ICML 2024.
> > **W3**: Figure 3 shows the prediction performance corresponding to different step lengths, but it lacks experimental results for some extreme cases, such as reducing the step length of exogenous/endogenous variables to 1.
>
> As per your request, we conduct experiments in some extreme cases. Concretely, we set the look-back length of exogenous/endogenous variables to 1 respectively and list the results as follows, where EN and EX denote the endogenous and exogenous series respectively.
>
> Short-term (MSE\|MAE)|NP|PJM|BE|FR|DE|AVG
> -|-|-|-|-|-|-
> |TimeXer|**0.236**\|**0.268**|**0.093**\|**0.192**|**0.379**\|**0.243**|**0.385**\|**0.208**|**0.440**\|**0.415**|**0.307**\|**0.265**
> |TimeXer-EN1|0.269\|0.296|0.110\|0.209|0.482\|0.324|0.442\|0.245|0.500\|0.453|0.361\|0.305
> |TimeXer-EX1|0.252\|0.277|0.106\|0.207|0.398\|0.255|0.415\|0.215|0.467\|0.429|0.328\|0.277
>
> We can find that TimeXer’s performance is relevant to the look-back length of the endogenous variable, deteriorating when historical information is severely limited. Additionally, the length of the exogenous variable also influences forecasting performance, with better results observed when more external information is available.
> > **W4**: It is recommended to explain more clearly in the methods and introduction sections why the global token is applied to the time series prediction task.
>
> Thanks for your suggestion, we will provide a more detailed description in $\underline{\text{Line 146-148 of main text}}$ to better explain the role of the global token in the revision: "Given the distinct roles of endogenous and exogenous variables in the prediction, TimeXer embeds them at different granularity. Directly combining patch-level endogenous tokens and variate-level exogenous tokens results in information misalignment. To address this, we introduce a learnable global token for each endogenous variable that serves as the macroscopic representation to interact with exogenous variables. This design helps bridge the causal information from the exogenous series to the endogenous temporal patches."
> > **Q1**: The influence of exogenous variables on endogenous variables is multi-grained, including aspects like periodicity, trend, noise, and stability. Does the Transformer have the capability to capture such multi-grained influences, or does the author think that more promising methods (such as LLMs) could be exploited to capture these associations?
>
> Thank you for your insightful question. Previous works, e.g. Autoformer, have attempted to use the seasonal-trend decomposition method to reveal the entangled temporal patterns in time series data and better capture underlying temporal dependencies. We think it will be reasonable to apply more elaborative designs (e.g. decomposition or LLMs) to the endogenous and exogenous series and leverage our proposed architecture to capture these multi-grained influences.
>
> Since we attempt to "unleash the potential of the canonical Transformer without modifying any component" ($\underline{\text{Lines 60-61 of Introduction}}$), we incorporate external information into TimeXer predictions through series-level dependencies, which works well. We would like to leave the exploration of multi-granularity effects in future work. As for the exploitation of LLMs, Reviewer 1zKu has a similar question, you can refer to our rebuttal to 1zKu Q1 for experiments on leveraging LLM.

---

> > ### Comment · Reviewer_Mh7v · 2024-08-10
> >
> > Thank you for your efforts in responding. Based on your replies, I have decided to raise my score to 6. Good luck.
> >
> > Here are some additional suggestions that may promote your manuscript's quality.  (1) how do you determine the exogenous variables, considering that the exogenous and endogenous variables in the traffic and electricity datasets are of the same type. (2) some key models designed for exogenous variables have been omitted, such as NARMAX, N-BEATSx, and TFT.

---

> > > ### Author Response · Authors · 2024-08-10
> > > **Thanks for your response and raising score.**
> > >
> > > Thank you for your prompt response and valuable comments on our paper, which have been of great help to us.
> > >
> > > Here is a brief answer to your concerns: (1) The great difference between exogenous and endogenous is whether they need to be predicted or not. We will provide more description of the definition in the camera-ready version. (2) Due to the page limit of the main text, we have compared TimeXer to N-BEATSx, and TFT in $\underline{\text{Appendix I.1}}$.
> > >
> > > Thanks again for your response and raising your score. We promise to follow your suggestion to improve our manuscript's quality in the revised revision.

---

### Official Review · Reviewer_r8UR · 2024-07-12

**Soundness:** 3
**Presentation:** 3
**Contribution:** 2
**Rating:** 6
**Confidence:** 4

**Summary:**

The paper presents TimeXer, a Transformer-based model for time series forecasting that integrates exogenous variables using innovative attention mechanisms. Unlike traditional models that either focus solely on endogenous variables or treat all variables equally, TimeXer integrates exogenous information using patch-wise self-attention and variate-wise cross-attention mechanisms.

**Strengths:**

1. Given the growing emphasis on multivariate time series forecasting in recent research, this paper distinguishes itself by focusing on exogenous variables. By integrating the strengths of recent studies and introducing novel ideas, it advances the field in a meaningful way.

2. Unlike other models that specialize in specific types of forecasting, TimeXer presents a versatile framework applicable to univariate forecasting, multivariate forecasting, and forecasting with exogenous variables. This adaptability makes it a comprehensive tool for a wide range of forecasting scenarios.

**Weaknesses:**

1. Methodological Gaps in Handling Practical Situations

   While the paper presents various practical situations involving exogenous variables, the methodology, specifically the variate embedding section, only proposes embedding through learning parameters. It lacks a detailed methodological approach to address these practical issues effectively.

2. Lack of Intuitive Examples for Exogenous Variable Utilization
   The paper does not provide intuitive examples of how well the model learns and uses exogenous variables. Including examples where TimeXer can effectively learn from other variables in scenarios where information is only available from external variables would significantly emphasize the model's necessity and practical relevance.

3. Lack of Causality Between Variable Similarity and Prediction Accuracy
   There is an inherent lack of causality between the similarity of shapes among variables and the accuracy of predictions. In Figure 5 (left), the paper demonstrates score differences when two series look similar versus when they do not. However, this does not convincingly show that the presence of similarly patterned time series aids in prediction accuracy.

**Questions:**

1. The paper would benefit from more intuitive examples demonstrating how TimeXer effectively learns and utilizes exogenous variables. Can you provide practical examples or case studies where TimeXer successfully learns from exogenous variables in scenarios where only external information is available?

2. In Figure 5 (left), the paper shows score differences when two series look similar versus when they do not, but it does not convincingly establish that the presence of similarly patterned time series aids in prediction accuracy. Can you provide more evidence or analysis to support the claim that similarity in variable patterns enhances prediction accuracy?

3. In the experiments, univariate forecasting with exogenous variables and multivariate forecasting were not distinctly separated. Why did you choose not to clearly differentiate between these two types of forecasting scenarios in your descriptions?

**Limitations:**

Limitations are discussed briefly.

---

> ### Author Rebuttal · Authors · 2024-08-06
>
> # Response to Reviewer r8UR
> We would like to sincerely thank Reviewer r8UR for providing a detailed review and insightful suggestions.
>
> > **W1**: Methodological Gaps in Handling Practical Situations
>
> As the reviewer mentioned, we focus on forecasting with exogenous variables and have provided a comprehensive analysis for practical forecasting scenarios in $\underline{\text{Lines 31-39 of main text}}$. Although the real-world scenarios are multifarious, our proposed TimeXer provides a **simple, flexible, and effective** method to tackle practical challenges, such as missing value or mismatch information.
>
> With our specially designed variate embedding layer (learned embedding and cross-attention module), TimeXer can naturally handle the mismatch series length and diverse information, which has been comprehensively verified in our experiments:
>
> (1) **Mismatch Length**: In $\underline{\text{Lines 255-256 and Figure 3 of main text}}$, we enlarge the length of exogenous and endogenous series respectively, resulting in unequal input lengths.
>
> (2) **Mismatch Frequency and Temporal Misaligned**: We conduct experiments on large-scale time series data ($\underline{\text{Figure 4 of main text}}$). As we stated in $\underline{\text{Lines 275-276 of main text}}$, the sampling frequencies of the endogenous and exogenous variables are 1h and 3h respectively. Also, the temperature, pressure, and wind meteorological indicators exist inherent temporally misaligned relation (temperature will change earlier, then pressure and finally wind).
>
> (3) **Missing Values**: In $\underline{\text{Section C of the Appendix}}$, we randomly mask the exogenous series to obtain cases with missing values.
>
> The above results verify that **our variate embedding design provides a simple and neat way to bridge Transformer to practical situations, whose flexibility and effectiveness should not be overlooked.**
>
> > **W2&Q1**: Lack of Intuitive Examples for Exogenous Variable Utilization
>
> Following your suggestion, we have included more showcases in $\underline{\text{Figure 1 of the Author Rebuttal PDF}}$.
>
> - We visually present the prediction results in two cases, with and without exogenous variables, to validate the role of exogenous variables.
> - As per your request in **Q1**, we add an extreme case where there is no historical information on endogenous series to explore whether TimeXer can learn from exogenous variables in scenarios where only external information is available.
> - In addition, we also add a special case where the prediction of exogenous variables is available to the model. This is a practical scenario for the $\underline{\text{EPF datasets}}$ where the exogenous variables are the day-ahead predictions of the source generation.
>
> We can observe that the inclusion of exogenous variables successfully enhances model performance, and when further exogenous variable predictions are utilized, the performance achieves the best.
>
>
> > **W3&Q2**: Lack of Causality Between Variable Similarity and Prediction Accuracy
>
> In our paper, we adopt $\underline{\text{Figure 5 of main text}}$ to demonstrate that the cross-attention module empowers TimeXer with special interpretability. We would like to highlight that the similarity we mentioned is not directly equivalent to "shape similarity"; rather, it is derived from the attention mechanism, which captures the intrinsic temporal patterns present within time series data. Intuitively, time series with similar shapes may exhibit shared temporal features, resulting in a higher similarity score. Thus, the most similar exogenous series learned by the attention mechanism may intuitively resemble the endogenous variable, but it cannot be said that all similar series will definitely contribute to the prediction.
>
> As per the reviewer's request, we conduct analysis studies on the causality between variable similarity and model performance. Technologically, we select the most similar and least similar variable to the endogenous variables through the learned attention map and further conduct experiments by removing this exogenous variable. The results are listed as follows, which can observe that series with larger similarity (learned by TimeXer) is more beneficial to the prediction.
>
> |Short-term(MSE\|MAE)|NP|PJM|BE|FR|DE|AVG|
> -|-|-|-|-|-|-
> | TimeXer    | **0.236** \| **0.268** | **0.093** \| **0.192** | **0.379** \| **0.243** | **0.385** \| **0.208** | **0.440** \| **0.415** | **0.307** \| **0.265** |
> |TimeXer-w/o Most Similar| 0.295 \| 0.299 | 0.096 \|0.194 | 0.385 \| 0.246 | 0.393 \| 0.210 | 0.483 \| 0.439 | 0.331 \|0.278 |
> |TimeXer-w/o Least Similar| 0.244 \| 0.273 | 0.093 \|0.192 | 0.382\| 0.242 | 0.387 \| 0.211 | 0.462 \| 0.422 | 0.313 \|0.268 |
>
>
> > **Q3**: In the experiments, univariate forecasting with exogenous variables and multivariate forecasting were not distinctly separated. Why did you choose not to clearly differentiate between these two types of forecasting scenarios in your descriptions?
>
> As stated in $\underline{\text{Line 193 of main text}}$, we find that **forecasting with exogenous variables can be a unified forecasting paradigm** that generalizes straightforwardly to multivariate forecasting by taking each variable as endogenous, the other variables are exogenous.
>
> To verify the effectiveness and generality of TimeXer, we conduct experiments under two different forecasting paradigms, namely short-term forecasting with exogenous variables on EPF benchmarks and long-term multivariate forecasting on well-established benchmarks. The experimental results for these two distinct forecasting tasks are given in $\underline{\text{Tables 2, 3 of main text}}$, respectively.

---

> > ### Comment · Reviewer_r8UR · 2024-08-13
> >
> > Thank you for your rebuttal and for addressing my concerns. While I’m maintaining my original score of weak acceptance, I appreciate the improvements and clarifications you’ve made. Your effort to refine the paper is evident, and I value the thoughtful response.

---

### Author Rebuttal · Authors · 2024-08-06

## Summary of Revisions and Global Response

We sincerely thank all the reviewers for their insightful reviews and valuable comments, which are instructive for us to improve our paper further.

In this paper, we dive into a practical forecasting setting in real-world applications, i.e. time series forecasting with exogenous variables. We propose TimeXer as a simple and general model that explicitly distinguishes between endogenous and exogenous variables through two different embedding modules, with an endogenous global token as a bridge in-between. With this design, we empower Transformer architecture to the inclusion of exogenous variables without architectural modifications. **Experimentally, TimeXer surpasses 9 advanced baselines in 12 well-established datasets in both short-term forecasting with exogenous variables and long-term multivariate forecasting with favorable efficiency and interpretability.**

The reviewers generally held positive opinions of our paper, in that the proposed method is "**novel**", "**well-thought-out**", **a versatile framework**", "**intuitively effective**", and "**advances the field in a meaningful way**", "**has a strong practical value**", "**has potential implications**"; the experiments are "**extensive**", "**comprehensive**", "**interesting and convincing**", "**carefully executed**" and demonstrate "**adaptability**", "**the effectiveness of the method across various datasets and experimental settings**".

The reviewers also raised insightful and constructive concerns. We made every effort to address all the concerns by providing detailed descriptions and requested results. Here is the summary of the major revisions:

- **Provide more intuitive examples for exogenous variable utilization (Review r8UR):** We visually present the prediction showcases in four different cases in the following PDF, including only using endogenous or exogenous series, using both of these two kinds of series and using the prediction values of exogenous series, to validate the role of exogenous variables.

- **Explain why our method uses the non-overlapping patch method (Reviewer Mh7v):** Following the reviewer's question, we conduct an ablation study on the effects of patching, we find that the non-overlapping has the lowest complexity with the optimal performance.

- **Explain how to extend TimeXer to multivariate forecasting tasks (Reviewer Mh7v, N2A5):** Following the reviewer's suggestion, we elucidate the rationale behind TimeXer's adaptability to multivariate prediction tasks.

- **Experiments under extreme cases (Reviewer Mh7v, 1zKu):** As per the reviewers' request, we have completed comprehensive ablation studies in five distinct scenarios, including reducing the step length of exogenous/endogenous variables to 1, setting the exogenous/endogenous variables to zeros, and setting the exogenous variables to random numbers. TimeXer exhibits adaptability across these diverse scenarios.

- **Add comparison with CrossGNN and FourierGNN (Reviewer 1zKu):** Following the reviewer's suggestion, we have compared the above two baselines in short-term forecasting benchmarks. Compared to these new baselines, TimeXer still performs best.

- **Explore the usage of LLMs in forecasting with exogenous variables (Reviewer 1zKu):** Following the reviewer's suggestion, we have designed a textual prompt to describe the exogenous variable information and tested whether the exploitation of large language models can improve the model's ability to understand exogenous variables at multiple granularities.

The valuable suggestions from reviewers are very helpful for us to revise the paper to a better shape. We'd be very happy to answer any further questions.

Looking forward to the reviewer's feedback.

#### **The mentioned materials are included in the following PDF file.**

- **Figure 1 (Reviewer r8UR)**: Intuitive showcase of the prediction results under different scenarios.
- **Table 1 (Reviewer 1zKu)**: Experimental results for W4 and Q2.

---

### Decision · Program_Chairs · 2024-09-25

**Decision:**

Accept (poster)

**Comment:**

This paper proposes a new time series forecasting model that uses exogenous variables in the prediction. The exogenous variables are valuable for forecasting but they are often ignored in the existing models. The proposed method unifies exogenous and endogenous variables using patch-wise self-attention and variate-wise cross-attention. The proposed method shows state-of-the-art performance on real-world forecasting benchmarks. The reviewers consider this work as novel and of strong practical value. The experimental studies in this paper are extensive, which makes the results trustable. I would like to recommend it for acceptance.